# INDIVIDUALLY FAIR GRADIENT BOOSTING

**Alexander Vargo**
Department of Mathematics
University of Michigan
ahsvargo@umich.edu

**Fan Zhang**
School of Information Science and Technology
ShanghaiTech University
zhangfan4@shanghaitech.edu.cn

**Mikhail Yurochkin**
IBM Research
MIT-IBM Watson AI Lab
mikhail.yurochkin@ibm.com

**Yuekai Sun**
Department of Statistics
University of Michigan
yuekai@umich.edu

## ABSTRACT

We consider the task of enforcing individual fairness in gradient boosting. Gradient boosting is a popular method for machine learning from tabular data, which arise often in applications where algorithmic fairness is a concern. At a high level, our approach is a functional gradient descent on a (distributionally) robust loss function that encodes our intuition of algorithmic fairness for the ML task at hand. Unlike prior approaches to individual fairness that only work with smooth ML models, our approach also works with non-smooth models such as decision trees. We show that our algorithm converges globally and generalizes. We also demonstrate the efficacy of our algorithm on three ML problems susceptible to algorithmic bias.

## 1 INTRODUCTION

In light of the ubiquity of machine learning (ML) methods in high-stakes decision-making and support roles, there is concern about ML models reproducing or even exacerbating the historical biases against certain groups of users. These concerns are valid: there are recent incidents in which algorithmic bias has led to dire consequences. For example, Amazon recently discovered its ML-based resume screening system discriminates against women applying for technical positions (Dastin, 2018).

In response, the ML community has proposed a myriad of formal definitions of algorithmic fairness. Broadly speaking, there are two types of fairness definitions: group fairness and individual fairness (Chouldechova & Roth, 2018). In this paper, we focus on enforcing individual fairness. At a high-level, the idea of individual fairness is the requirement that a fair algorithm should treat similar individuals similarly. For a while, individual fairness was overlooked in favor of group fairness because there is often no consensus on which users are similar for many ML tasks. Fortunately, there is a flurry of recent work that addresses this issue (Ilvento, 2019; Wang et al., 2019; Yurochkin et al., 2020; Mukherjee et al., 2020a). In this paper, we assume there is a fair metric for the ML task at hand and consider the task of individually fair gradient boosting.

Gradient boosting, especially gradient boosted decision trees (GBDT), is a popular method for tabular data problems (Chen & Guestrin, 2016). Unfortunately, existing approaches to enforcing individual fairness are either not suitable for training non-smooth ML models (Yurochkin et al., 2020) or perform poorly with flexible non-parametric ML models. We aim to fill this gap in the literature. Our main contributions are:

1. We develop a method to enforce individual fairness in gradient boosting. Unlike other methods for enforcing individually fairness, our approach handles non-smooth ML models such as (boosted) decision trees.
2. We show that the method converges globally and leads to ML models that are individually fair. We also show that it is possible to *certify* the individual fairness of the models *a posteriori*.
3. We show empirically that our method preserves the accuracy of gradient boosting while improving widely used group and individual fairness metrics.

## 2 ENFORCING INDIVIDUAL FAIRNESS IN GRADIENT BOOSTING

Consider a supervised learning problem. Let $\mathcal{X} \subset \mathbb{R}^d$ be the input space and $\mathcal{Y}$ be the output space. To keep things simple, we assume $\mathcal{Y} = \{0, 1\}$, but our method readily extends to multi-class classification problems. Define $\mathcal{Z} = \mathcal{X} \times \{0, 1\}$. We equip $\mathcal{X}$ with a *fair metric* $d_x$ that measures the similarity between inputs. The fair metric is application specific, and we refer to the literature on fair metric learning (Ilvento, 2019; Wang et al., 2019; Yurochkin et al., 2020) for ways of picking the fair metric. Our goal is to learn an ML model $f : \mathcal{X} \to \{0, 1\}$ that is individually fair. Formally, we enforce distributionally robust fairness (Yurochkin et al., 2020), which asserts that an ML model has similar accuracy/performance (measured by the loss function) on similar samples (see Definition 2.1).

One way to accomplish this is adversarial training (Yurochkin et al., 2020; Yurochkin & Sun, 2020). Unfortunately, adversarial training relies on the smoothness of the model (with respect to the inputs), so it cannot handle non-smooth ML models (*e.g.* decision trees). We address this issue by considering a restricted adversarial cost function that only searches over the training examples (instead of the entire input space) for similar examples that reveal violations of individual fairness. As we shall see, this restricted adversarial cost function is amenable to functional gradient descent, which allows us to develop a gradient boosting algorithm.

### 2.1 ENFORCING INDIVIDUAL FAIRNESS WITH A RESTRICTED ADVERSARIAL COST FUNCTION

We first review how to train individually fair ML models with adversarial training (see Yurochkin et al. (2020) for more details) to set up notation and provide intuition on how adversarial training leads to individual fairness. Let $\mathcal{F}$ be a set of ML models, let $\ell : \mathcal{Y} \times \mathcal{Y} \to \mathbb{R}$ be a smooth loss function (see Section 3 for concrete assumptions on $\ell$) that measures the performance of an ML model, and let $\mathcal{D} = \{(x_i, y_i)\}_{i=1}^n$ be a set of training data. Define the transport cost function

$$c((x_1, y_1), (x_2, y_2)) \triangleq d_x^2(x_1, x_2) + \infty \cdot \mathbf{1}_{\{y_1 \neq y_2\}}. \tag{2.1}$$

We see that $c((x_1, y_1), (x_2, y_2))$ is small iff $x_1$ and $x_2$ are similar (in the fair metric $d_x$) and $y_1 = y_2$. In other words, $c$ is small iff two similar examples are assigned the same output. Define the optimal transport distance $W$ (with transport cost $c$) on probability distributions on $\mathcal{Z}$:

$$W(P_1, P_2) \triangleq \inf_{\Pi \in C(P_1, P_2)} \int_{\mathcal{Z} \times \mathcal{Z}} c(z_1, z_2) d\Pi(z_1, z_2),$$

where $C(P_1, P_2)$ is the set of couplings between $P_1$ and $P_2$ (distributions on $\mathcal{Z} \times \mathcal{Z}$ whose marginals are $P_1$ and $P_2$). This optimal transport distance lifts the fair metric on (points in) the sample space to distributions on the sample space. Two distributions are close in this optimal transport distance iff they assign mass to similar areas of the sample space $\mathcal{Z}$. Finally, define the adversarial risk function

$$L_r(f) \triangleq \sup_{P: W(P, P_*) \leq \epsilon} \mathbb{E}_P[\ell(f(X), Y)], \tag{2.2}$$

where $P_*$ is the data generating distribution and $\epsilon > 0$ is a small tolerance parameter. The adversarial risk function looks for distributions on the sample space that (i) are similar to the data generating distribution and (ii) increases the risk of the ML model $f$. This reveals differential performance of the ML model on similar samples. This search for differential performance is captured by the notion of distributionally robust fairness:

**Definition 2.1** (distributionally robust fairness (DRF) (Yurochkin et al., 2020))**.** *An ML model* $h : \mathcal{X} \to \mathcal{Y}$ *is* $(\epsilon, \delta)$*-distributionally robustly fair (DRF) WRT the fair metric* $d_x$ *iff*

$$\sup_{P: W(P, P_n) \leq \epsilon} \int_{\mathcal{Z}} \ell(z, h) dP(z) \leq \delta. \tag{2.3}$$

In light of the preceding developments, a natural cost function for training individually fair ML models is the adversarial cost function:

$$L_e(f) \triangleq \sup_{P: W(P, P_n) \leq \epsilon} \mathbb{E}_P[\ell(f(X), Y)], \tag{2.4}$$

where $P_n$ is the empirical distribution of the training data. This is the empirical counterpart of (2.2), and it works well for training smooth ML models (Yurochkin et al., 2020). Unfortunately, (2.4) is hard to evaluate for non-smooth ML models: it is defined as the optimal value of an optimization problem, but the gradient $\partial_x \ell(f(x), y)$ is not available because the ML model $f$ is non-smooth.

To circumvent this issue, we augment the support of the training set and restrict the supremum in (2.4) to the augmented support. Define the augmented support set $\mathcal{D}_0 \triangleq \{(x_i, y_i), (x_i, 1 - y_i)\}_{i=1}^n$ and the restricted optimal transport distance $W_\mathcal{D}$ between distributions *supported on* $\mathcal{D}_0$:

$$W_\mathcal{D}(P_1, P_2) \triangleq \inf_{\Pi \in C_0(P_1, P_2)} \int_{\mathcal{Z} \times \mathcal{Z}} c(z_1, z_2) d\Pi(z_1, z_2),$$

where $C_0(P_1, P_2)$ is the set of distributions supported on $\mathcal{D}_0 \times \mathcal{D}_0$ whose marginals are $P_1$ and $P_2$. We consider the restricted adversarial cost function

$$L(f) \triangleq \sup_{P: W_\mathcal{D}(P, P_n) \leq \epsilon} \mathbb{E}_P[\ell(f(X), Y)], \tag{2.5}$$

where $\epsilon > 0$ is a small tolerance parameter. The interpretation of (2.5) is identical to that of (2.4): it searches for perturbations to the training examples that reveal differential performance in the ML model. On the other hand, compared to (2.4), the supremum in (2.5) is restricted to distributions supported on $\mathcal{D}_0$. This allows us to evaluate (2.5) by solving a (finite-dimensional) linear program (LP). As we shall see, this LP depends only on the loss values $\ell(f(x_i), y_i)$ and $\ell(f(x_i), 1 - y_i)$, so it is possible to solve the LP efficiently even if the ML model $f$ is non-smooth. *This is the key idea in this paper.*

Before delving into the details, we note that the main drawback of restricting the search to distributions supported on $\mathcal{D}_0$ is reduced power to detect differential performance. If the ML model exhibits differential performance between two (similar) areas of the input space but only one area is represented in the training set, then (2.4) will detect differential performance but (2.5) will not. Augmenting the support set with the points $\{(x_i, 1 - y_i)\}_{i=1}^n$ partially alleviates this issue (but the power remains reduced compared to (2.4)). This is the price we pay for the broader applicability of (2.5).

## 2.2 FUNCTIONAL GRADIENT DESCENT ON THE RESTRICTED ADVERSARIAL COST FUNCTION

Gradient boosting is functional gradient descent (Friedman, 2001), so a key step in gradient boosting is evaluating $\frac{\partial L}{\partial \hat{y}}$, where the components of $\hat{y} \in \mathbb{R}^n$ are $\hat{y}_i \triangleq f(x_i)$. By Danskin's theorem, we have

$$\frac{\partial L}{\partial \hat{y}_i} = \frac{\partial}{\partial f(x_i)} \big[ \sup_{P: W_\mathcal{D}(P, P_n) \leq \epsilon} \mathbb{E}_P[\ell(f(x_i), y_i)] \big] = \sum_{y \in \mathcal{Y}} \frac{\partial}{\partial f(x_i)} \big[ \ell(f(x_i), y)) \big] P^*(x_i, y), \tag{2.6}$$

where $P^*$ is a distribution that attains the supremum in (2.5). We note that there is no need to differentiate through the ML model $f$ in (2.6), so it is possible to evaluate the functional gradient for non-smooth ML models. It remains to find $P^*$. We devise a way of finding $P^*$ by solving a linear program.

We start with a simplifying observation: if $c(z_i, z_j) = \infty$ for any $z_i \in \mathcal{D}_0$ and $z_j \in \mathcal{D}$, then any weight at $z_j$ cannot be transported to $z_i$. Thus, we will only focus on the pairs $(z_i, z_j) \in \mathcal{D}_0 \times \mathcal{D}$ with $c(z_i, z_j) < \infty$. Let $C \in \mathbb{R}^{n \times n}$ be the matrix with entries given by $C_{i,j} = c((x_i, y_j), (x_j, y_j)) = d_x^2(x_i, x_j)$. We also define the class indicator vectors $y^1, y^0 \in \{0, 1\}^n$ by

$$y_j^1 = \begin{cases} 1 & : y_j = 1 \\ 0 & : y_j = 0 \end{cases} \quad \text{and} \quad y^0 = \mathbf{1}_n - y^1. \tag{2.7}$$

For any distribution $P$ on $\mathcal{D}_0$, let $P_{i,k} = P(\{(x_i, k)\})$ for $k \in \{0, 1\}$. Then, the condition that $W_\mathcal{D}(P, P_n) \leq \epsilon$ is implied by the existence of a matrix $\Pi$ such that

1. $\Pi \in \Gamma$ with $\Gamma = \{\Pi | \Pi \in \mathbb{R}_+^{n \times n}, \langle C, \Pi \rangle \leq \epsilon, \Pi^T \cdot \mathbf{1}_n = \frac{1}{n} \mathbf{1}_n\}$.
2. $\Pi \cdot y^1 = (P_{1,1}, \ldots, P_{n,1})$, and $\Pi \cdot y^0 = (P_{1,0}, \ldots, P_{n,0})$.

Further define the matrix $R \in \mathbb{R}^{n \times n}$ by $R_{ij} = \ell(f(x_i), y_j)$ - this is the loss incurred if point $j$ with label $y_j$ is transported to point $i$. With this setup, given the current predictor $f$, we can obtain a solution $\Pi^*$ to the optimization as the solution to the linear program (in $n^2$ variables)

$$\Pi^* \in \arg \max_{\Pi \in \Gamma} \langle R, \Pi \rangle. \tag{2.8}$$

Then the optimal distribution $P^*$ on $\mathcal{D}_0$ is given by $P^*(\{(x_i, k)\}) = (\Pi^* \cdot y^k)_i$. An outline of the full gradient boosting procedure is provided in Algorithm 1.

It is important to note that we have made no assumptions about the class of candidate predictors $\mathcal{F}$ in finding the optimal transport map $\Pi^*$ in (2.8). In particular, $\mathcal{F}$ can contain discontinuous functions - for example, decision trees or sums of decision trees. This allows us to apply this fair gradient boosting algorithm to any class $\mathcal{F}$ of base classifiers.

---

**Algorithm 1** Fair gradient boosting

---

1: **Input**: Labeled training data $\{(x_i, y_i)\}_{i=1}^n$; class of weak learners $\mathcal{H}$; initial predictor $f_0$; search radius $\epsilon$; number of steps $T$; sequence of step sizes $\alpha^{(t)}$; fair metric $d_x$ on $\mathcal{X}$
2: Define the matrix $C$ by $C_{i,j} \leftarrow d_x^2(x_i, x_j)$.
3: **for** $t = 0, 1, \ldots, T-1$ **do**
4:     Define the matrix $R_t$ by $(R_t)_{ij} = \ell(f_t(x_i), y_j)$
5:     Find $\Pi_t^* \in \arg\max_{\Pi \in \Gamma} \langle R_t, \Pi \rangle$; and set $P_{t+1}(x_i, k) \leftarrow (\Pi_t^* \cdot y^k)_i$
6:     Fit a base learner $h_t \in \mathcal{H}$ to the set of pseudo-residuals $\{\frac{\partial L}{\partial f_t(x_i)}\}_{i=1}^n$ (see (2.6)).
7:     Let $f_{t+1} = f_t + \alpha_t h_t$.
8: **end for**
9: **return** $f_T$

---

## 2.3 RELATED WORK

**Enforcing individual fairness.** There is a line of work that seeks to improve fairness guarantees for individuals by enforcing group fairness with respect to many (possibly overlapping) groups (Hébert-Johnson et al., 2017; Kearns et al., 2017; Creager et al., 2019; Kearns et al., 2019; Kim et al., 2019). There is another line of work on enforcing individual fairness without access to a fair metric (Gillen et al., 2018; Kim et al., 2018; Jung et al., 2019; Kearns et al., 2019). This line of work circumvents the need for a fair metric by assuming the learner has access to an oracle that provides feedback on violations of individual fairness. Our work is part of a third line of work on enforcing individual fairness that assumes access to a fair metric (Yurochkin et al., 2020; Yurochkin & Sun, 2020). The applicability of these methods has broadened thanks to recent work on the precursor task of learning the fair metric from data (Ilvento, 2019; Lahoti et al., 2019; Wang et al., 2019; Mukherjee et al., 2020b). Unfortunately, these methods are limited to training smooth ML models.

**Adversarial training and distributionally robust optimization.** Our approach to fair training is also similar to adversarial training (Goodfellow et al., 2014; Madry et al., 2017), which hardens ML models against adversarial attacks. There are two recent papers on fitting robust non-parametric classifiers (including decision trees) (Yang et al., 2019; Chen et al., 2019) but neither are applicable to enforcing individual fairness in gradient boosting. Our approach is also an instance of distributionally robust optimization (DRO) (Blanchet et al., 2016; Duchi & Namkoong, 2016; Esfahani & Kuhn, 2015; Lee & Raginsky, 2017; Sinha et al., 2017; Hashimoto et al., 2018).

Before moving on, we remark that the key idea in this paper cannot be applied to adversarial training of non-smooth ML models. The goal of adversarial training is to make ML model robust to adversarial examples that are not in the training set. The restriction to the augmented support set in (2.5) precludes such adversarial examples, so training with (2.5) does not harden the model against adversarial examples. On the other hand, as long as the training set is diverse enough (see Assumption 3.3 for a rigorous condition to this effect), it is possible to reveal differential performance in the ML model by searching over the augmented support set and enforce individual fairness.

## 3 THEORETICAL RESULTS

We study the convergence and generalization properties of fair gradient boosting (Algorithm 1). The optimization properties are standard: fair gradient boosting together with a line search strategy converges globally. This is hardly surprising in light of the global convergence properties of line search methods (see (Nocedal & Wright, 2006, Chapter 3)), so we defer this result to Appendix A.1.

The generalization properties of fair gradient boosting are less standard. We start by stating the assumptions on the problem. The first two assumptions are standard in the DRO literature (see Lee & Raginsky (2017) and Yurochkin et al. (2020)).

**Assumption 3.1** (boundedness of input space). $\operatorname{diam}(\mathcal{X}) < \infty$

**Assumption 3.2** (regularity of loss function). *(i) $\ell$ is bounded: $0 \le \ell(f, z) \le B \; \forall \, f \in \mathcal{F}, \, z \in \mathcal{Z}$.*
*(ii) Let $\mathcal{L} \triangleq \{\ell(f, \cdot) : f \in \mathcal{F}\}$ denote the class of loss functions. $\mathcal{L}$ is $\omega_2$-Lipschitz with respect to $d_x$:*
    $|\ell(f, (x_1, y)) - \ell(f, (x_2, y))| \le \omega_2 d_x(x_1, x_2)$ *for all $x_1, x_2 \in \mathcal{X}, \, y \in \mathcal{Y}$ and $f \in \mathcal{F}$.*

Additionally, the support of the data generating distribution $P_*$ should cover the input space. Otherwise, the training data may miss areas of the input space, which precludes detecting differential treatment in these areas. Similar conditions appear in the non-parametric classification literature, under the name *strong density condition* (Audibert & Tsybakov, 2007).

**Assumption 3.3.** *Let $B_{d_x}(r, x_*) = \{x \in \mathcal{X} : d_x(x, x_*) < r\}$ be the $d_x$-ball of radius $r$ around $x_*$ in $\mathcal{X}$. There are constants $\delta > 0$ and $d$ such that $P_*(B_{d_x}(r, x) \times \mathcal{Y}) \geq \delta r^d$ for any $r < 1$.*

The lower bound $\delta r^d$ in Assumption 3.3 is motivated by the volume of the Euclidean ball of radius $r$ in $\mathbb{R}^d$ being proportional to $r^d$. For a bounded input space $\mathcal{X}$, Assumption 3.3 implies the probability mass assigned by $P_*$ to small balls is always comparable (up to the small constant $\delta$) to the probability mass assigned by the uniform distribution on $\mathcal{X}$. We note that this assumption is close to the goal of Buolamwini & Gebru (2018) in their construction of the Pilot Parliaments Benchmark data.

**Theorem 3.4.** *Under Assumptions 3.1, 3.2, and 3.3, we have*

$$\sup_{f \in \mathcal{F}} |L(f) - L_r(f)| \lesssim_P \frac{1}{n^{1/(2d)}} \left( \omega_2 + \frac{2\omega_2 \text{diam}(\mathcal{X})}{\sqrt{\epsilon}} \right) + \frac{1}{\sqrt{n}} \tag{3.1}$$

The first term on the right side of (3.1) is the discrepancy between $L$ and $L_e$, while the second term is the discrepancy between $L_e$ and $L_r$ (recall (2.2) and (2.4)). The second term is well-studied in the DRO literature (Lee & Raginsky, 2017; Yurochkin et al., 2020), so we focus on the first term, which captures the effect of restricting the search space in the supremum in (2.5) to the augmented support set $\mathcal{D}_0$. We see the curse of dimensionality in the slow $\frac{1}{n^{1/(2d)}}$ convergence rate, but this is unavoidable. For the restricted search for differential performance on $\mathcal{D}_0$ to emulate the search on $\mathcal{X}$, the size of $\mathcal{D}_0$ must grow exponentially with the dimension $d$. This leads to the exponential dependence on $d$ in the first term.

We note that it is possible for the dimension $d$ in Assumption 3.3 to be smaller than $\dim(\mathcal{X})$. Intuitively, the fair metric $d_x$ ignores variation in the inputs due to the sensitive attributes. This variation is usually concentrated in a low-dimensional set because there are few sensitive attributes in most ML tasks. Thus the metric effectively restricts the search to this low-dimensional space, so the effective dimension of the search is smaller. In such cases, the convergence rate will depend on the (smaller) effective dimension of the search.

One practical consequence of Theorem 3.4 is it is possible to *certify a posteriori* that a (non-smooth) ML model is individually fair by checking the empirical performance gap

$$L(f) - \frac{1}{n} \sum_{i=1}^n \ell(f(x_i), y_i). \tag{3.2}$$

As long as the (non-adversarial) empirical risk converges to its population value, Theorem 3.4 implies

$$\sup_{f \in \mathcal{F}} |L(f) - \frac{1}{n} \sum_{i=1}^n \ell(f(x_i), y_i) - (L_r(f) - \mathbf{E}_{P_*}[\ell(f(X), Y)])| \xrightarrow{\text{P}} 0. \tag{3.3}$$

In other words, the (empirical) performance gap $L(f) - \frac{1}{n} \sum_{i=1}^n \ell(f(x_i), y_i)$ generalizes. Thus it is possible for practitioners to certify the worst-case performance differential of an ML model (up to an error term that vanishes in the large sample limit) by evaluating (3.2).

# 4 SCALABLE FAIR GRADIENT BOOSTING

A key step in the fair gradient boosting Algorithm 1 is finding the worst-case distribution $P^*$. This entails solving a linear program (LP) in $n^2$ variables ($n$ is the size of the training set). It is possible to use an off-the-shelf LP solver to find $P^*$, but solving an LP in $n^2$ variables per iteration does not scale to modern massive datasets. In this section, we appeal to duality to derive a stochastic optimization approach to finding the worst-case distribution. At a high level, the approach consists of two steps:

1. Solve the dual of (2.8). The dual is a univariate optimization problem, and it is amenable to stochastic optimization (see (B.14)). Computational complexity is $O(n)$.
2. Reconstruct the primal optimum from the dual optimum. Computational complexity is $O(n^2)$.

We see that the computational complexity of this step is $O(n^2)$, which is much smaller than the complexity of solving an LP in $n^2$ variables. This two-step approach is similar to the dual approach

to solving (unrestricted) DRO problems. Due to space constraints, we summarize the two steps in Algorithm 2 and defer derivations to Appendix B. We also include in Appendix B an entropic regularized version of the approach that sacrifices exactness for additional computational benefits.

We remark that it is possible to achieve further speedups by exploiting the properties of the fair metric. We start by observing that the cost of recovering the primal solution (step 10 in Algorithm 2) dominates the computational cost of Algorithm 2. Recall the search for differential performance is restricted by the fair metric to a small set of points. This implies the argmax in step 10 of Algorithm 2 is restricted to a few points that are similar to $x_j$ in the fair metric. This can be seen from the corresponding argmax expression: if $C_{ij}$ is large, $i$ is unlikely to be a solution to argmax. The neighbors of each point can be computed in advance and re-used during training. This further reduces the cost of recovering the primal solution from $O(n^2)$ to $O(nm)$, where $m$ is the maximum number of neighbors of a point. With this heuristic it should be possible to train our algorithm whenever the vanilla GBDT is feasible.

We now state the full fair gradient boosting algorithm that combines the framework presented in Section 2.1 with the preceding stochastic optimization approach to evaluating the worst-case distribution. We refer to the GBDT method as TreeBoost; this can be replaced with any GBDT algorithm. In every boosting step in Algorithm 3, we find the optimal transport map $\Pi^*$ using Algorithm 2. We then boost for one step using the GBDT training methods on the augmented data set $\mathcal{D}_0$ weighted according to the distribution given by $P(\{x_i, k\}) = (\Pi^* \cdot y^k)_i$. There are multiple hyperparameters that can be tweaked in the GBDT model (e.g. the maximum depth of the trees in $\mathcal{F}$); we represent this by including a list of GBDT parameters $\rho$ as an input to Algorithm 3.

---

**Algorithm 2** SGD to find optimal dual variable $\eta^*$ and approximate $\Pi^*$

---

1: **Input**: Initial $\eta_1 > 0$; cost matrix $C$; loss matrix $R$; tolerance $\epsilon$; batch size $B$; step sizes $\alpha_t > 0$.
2: **repeat**
3:     Sample indices $j_1, \ldots, j_B$ uniformly at random from $\{1, \ldots n\}$.
4:     Let $R_t \leftarrow$ columns $j_1, \ldots, j_B$ of $R$. Let $C_t \leftarrow$ columns $j_1, \ldots, j_B$ of $C$. $\{R_t, C_t \in \mathbb{R}^{n \times B}\}$
5:     Let $w_t(\eta) \leftarrow \eta \cdot \epsilon + \frac{1}{B} \sum_{j=1}^{B} \max_i R_{ij} - \eta C_{ij}$
6:     $\eta_{t+1} \leftarrow \max\{0, \eta_t - \alpha_t \frac{d}{d\eta} w_t(\eta)\}$
7: **until** converged
8: Set $\Pi$ be an $n \times n$ matrix of zeros; $\eta^*$ is the final value from above.
9: **for** $j = 0$ to $n - 1$ **do**
10:     Choose $t \in \arg\max_i R_{ij} - \eta^* C_{ij}$ and set $\Pi_{tj} = \frac{1}{n}$.
11: **end for**
12: **return** $\Pi$

---

**Algorithm 3** Fair gradient boosted trees (BuDRO)

---

1: **Input**: Data $\mathcal{D} = \{(x_i, y_i)\}_{i=1}^{n}$; perturbation budget $\epsilon$; loss function $\ell$; fair metric $d_x$ on $\mathcal{X}$; number of boosting steps $T$; GBDT parameters $\rho$, batch size $B$
2: Let $\mathcal{D}_0 = \{(x_i, 0)\}_{i=1}^{n} \cup \{(x_i, 1)\}_{i=1}^{n}$ and define $C$ by $C_{ik} \leftarrow d_x(x_i, x_k)$.
3: Let $f_0 = \text{TreeBoost.Train}(\rho, \text{data} = \mathcal{D}_0, \text{Steps} = 1)$ {Run one step of plain boosting}
4: **for** $t = 0$ to $T - 1$ **do**
5:     Define $R$ by $R_{ij} = \ell(f_t, (x_i, y_j))$.
6:     Construct $\Pi_t$ following Algorithm 2 with inputs $C, R, \epsilon$, and $B$
7:     Let $w_t$ be the concatenation of $\Pi_t \cdot y^0$ and $\Pi_t \cdot y^1$.
8:     Let $f_{t+1} \leftarrow \text{TreeBoost.Train}(\rho, f_t, \text{data} = \mathcal{D}_0, \text{weights} = w_t, \text{Steps} = 1)$.
9: **end for**
10: **Return** $f_T$.

---

## 5 EXPERIMENTS

We apply BuDRO (Algorithm 3) to three data sets popular in the fairness literature. Full descriptions of these data sets and a synthetic example are included in Appendix C. Timing information is also found in Appendix C.6. BuDRO uses the XGBoost algorithm (Chen & Guestrin, 2016) for the GBDT

method, and $\ell$ is the logistic loss. These experiments reveal that BuDRO is successful in enforcing individual fairness while achieving high accuracy (leveraging the power of GBDTs). We also observe improvement of the group fairness metrics.

**Fair metric.** Recall that a practical implementation of our algorithm requires a fair metric. We consider the procedure from Yurochkin et al. (2020) to learn the fair metric from data. They propose a metric of the form $d_x^2(x_1, x_2) = \langle x_1 - x_2, Q(x_1 - x_2) \rangle$, where $Q$ is the projecting matrix orthogonal to some sensitive subspace. This sensitive subspace is formed by taking the span of the vectors orthogonal to decision boundaries of linear classifiers fitted to predict a (problem specific) set of protected attributes (e.g. gender, race or age). The idea behind this procedure is that the sensitive subspace captures variation in the data due to protected information of the individuals. A fair metric should treat individuals that only differ in their protected information similarly, i.e. a distance between a pair of individuals that only differ by a component in the sensitive subspace should be 0.

**Comparison methods.** We consider other possible implementations of fair GBDT given existing techniques in the literature. As mentioned previously, due to the non-differentiability of trees, the majority of other individual fairness methods are not applicable to the GBDT framework. For this reason, we limit our analysis to fair data preprocessing techniques before applying a vanilla GBDT method. We report the results when considering two preprocessing techniques: **project** that eliminates the protected attributes and projects out the sensitive subspace used for the fair metric construction (Yurochkin et al., 2020), and **reweigh** that balances the representations of protected groups by assigning different weights to the individuals (Kamiran & Calders, 2011).

**Evaluation metrics.** To evaluate the individual fairness of each method without appealing to the underlying fair metric (i.e. to avoid giving our method and project an unfair advantage at test time, and to verify generalization properties of the fair metric used for training), we report data-specific *consistency* metrics. Specifically, for each data set, we find a set of attributes that are not explicitly protected but are correlated with the protected attribute (*e.g.* is_husband or is_wife when gender is protected) and vary these attributes to create artificial counterfactual individuals. Such counterfactual individuals are intuitively similar to the original individuals and classifier output should be the same for all counterfactuals to satisfy individual fairness. We refer to classification consistency as the frequency of a classifier changing its predictions on the aforementioned counterfactuals.

We also examine group fairness for completeness. We consider the group fairness metrics introduced in De-Arteaga et al. (2019). Specifically, we compute the differences in TPR and TNR for each protected attribute, and report the maximum ($\text{GAP}_{\text{Max}}$) and the root-mean-squared statistics ($\text{GAP}_{\text{RMS}}$) of these gaps. See Appendix C for a full description of all comparison metrics.

## 5.1 GERMAN CREDIT

The German credit data set (Dua & Graff, 2017) contains information from 1000 individuals; the ML task is to label the individuals as good or bad credit risks. We treat age as the protected attribute in the German data set. The age feature is not binary; this precludes the usage of fair methods that assume two protected classes (including the reweighing preprocessing technique). In the United States, it is unlawful to make credit decisions based on the age of the applicant; thus, there are distinct legal reasons to be able to create a classifier that does not discriminate based on age.

For the fair metric, we construct the sensitive subspace by fitting ridge regression on age and augment it with an indicator vector for the age coordinate (see Appendix C.3). This subspace is also used for data preprocessing with the project baseline. For the individual fairness consistency evaluation we consider varying a personal status feature that encodes both gender and marital status (S-cons).

Table 1: German credit: average results over 10 splits into 80% training and 20% test data.

| Method | BAcc | Status cons | Age gaps $\text{GAP}_{\text{Max}}$ | $\text{GAP}_{\text{RMS}}$ |
|---|---|---|---|---|
| BuDRO | .715 | **.974** | **.185** | .151 |
| Baseline | **.723** | .920 | .310 | .241 |
| Project | .698 | .960 | .188 | **.144** |
| Baseline NN | .687 | .826 | .234 | .179 |

We present the results in Table 1 (see Table 5 for error bars). To compare classification performance we report balanced accuracy due to class imbalance in the data. The baseline (GBDTs with XGBoost) is the most accurate and significantly outperforms a baseline neural network (NN). The BuDRO method has the highest individual fairness (S-cons) score while maintaining a high accuracy and improved group fairness metrics. Preprocessing by projecting out the sensitive subspace is not as effective as BuDRO in improving individual fairness and also can negatively impact the performance.

## 5.2 ADULT

The Adult data set (Dua & Graff, 2017) is another common benchmark in the fairness literature. The task is to predict if an individual earns above or below $50k per year. We follow the experimental setup and comparison metrics from the prior work on individual fairness (Yurochkin et al., 2020) studying this data. Individual fairness is quantified with two classification consistency measures: one with respect to a relationship status feature (S-cons) and the other with respect to the gender and race (GR-cons) features. The sensitive subspace is learned via logistic regression classifiers for gender and race and is augmented with unit vectors for gender and race. Note that the project preprocessing baseline is guaranteed to have a perfect GR-cons score since those features are explicitly projected out; it is interesting to assess if it generalizes to S-cons, however.

Table 2: Adult: average results over 10 splits into 80% training and 20% test data. NN, SenSR and Adversarial Debiasing (Zhang et al., 2018) numbers are from Yurochkin et al. (2020).

| Method | BAcc | Individual fairness | | Gender gaps | | Race gaps | |
|---|---|---|---|---|---|---|---|
| | | S-cons | GR-cons | $\mathrm{GAP}_{\mathrm{Max}}$ | $\mathrm{GAP}_{\mathrm{RMS}}$ | $\mathrm{GAP}_{\mathrm{Max}}$ | $\mathrm{GAP}_{\mathrm{RMS}}$ |
| BuDRO | .815 | **.944** | .957 | .146 | .114 | .083 | .072 |
| Baseline | **.844** | .942 | .913 | .200 | .166 | .098 | .082 |
| Project | .787 | .881 | **1** | **.079** | .069 | .064 | .050 |
| Reweigh | .784 | .853 | .949 | .131 | .093 | **.056** | **.043** |
| Baseline NN | .829 | .848 | .865 | .216 | .179 | .105 | .089 |
| SenSR | .789 | .934 | .984 | .087 | **.068** | .067 | .055 |
| Adv. Deb. | .815 | .807 | .841 | .110 | .082 | .078 | .070 |

The results are in Table 2 (see Table 7 for error bars). The baseline GBDT method is again the most accurate; it produces poor gender gaps, however. BuDRO is less accurate than the baseline, but the gender gaps have shrunken considerably, and both the S-cons and GR-cons are very high. Project and reweighing produce the best group fairness results; however, their S-cons values are worse than the baseline (representing violations of individual fairness), and they also result in significant accuracy reduction.

Comparing to the results from Yurochkin et al. (2020), BuDRO is slightly less accurate than the baseline NN, but it improves on all fairness metrics. BuDRO matches the accuracy of adversarial debiasing and greatly improves the individual fairness results there. Finally, BuDRO improves upon the accuracy of SenSR while maintaining similar individual fairness results. We present additional studies of the trade-off between accuracy and fairness in Figure 3 of Appendix C.4.1.

Overall, this and the preceding experiment provide empirical evidence that BuDRO trains individually fair classifiers while still obtaining high accuracy due to the power of GBDT methods.

## 5.3 COMPAS

We study the COMPAS recidivism prediction data set (Larson et al., 2016). The task is to predict whether a criminal defendant would recidivate within two years. We consider `race` (Caucasian or not-Caucasian) and `gender` (binary) as protected attributes and use them to learn the sensitive subspace similarly to previous experiments. Due to smaller data set size, instead of Algorithm 2, we considered an entropy-regularized solution to (2.8) presented in Appendix B. The individual fairness consistency measures are gender consistency (G-cons) and race-consistency (R-cons).

The results are collected in Table 3 (see Table 9 for error bars). COMPAS is a data set where NNs outperform GBDT in terms of accuracy, but this results in poor group and individual fairness measurements. A NN trained with SenSR shows similar accuracy to BuDRO, but worse performance

Table 3: COMPAS: average results over 10 splits into 80% training and 20% test data.

| Method | Acc | Individual fairness | | Gender gaps | | Race gaps | |
|---|---|---|---|---|---|---|---|
| | | G-cons | R-cons | $GAP_{Max}$ | $GAP_{RMS}$ | $GAP_{Max}$ | $GAP_{RMS}$ |
| BuDRO | 0.652 | **1.000** | **1.000** | **0.099** | **0.124** | 0.125 | 0.145 |
| Baseline | 0.677 | 0.944 | 0.981 | 0.180 | 0.223 | 0.215 | 0.258 |
| Project | 0.671 | 0.874 | **1.000** | 0.150 | 0.190 | 0.185 | 0.230 |
| Reweigh | 0.666 | 0.788 | 0.813 | 0.207 | 0.245 | **0.069** | **0.092** |
| Baseline NN | **0.682** | 0.841 | 0.908 | 0.246 | 0.282 | 0.228 | 0.258 |
| SenSR | 0.652 | 0.977 | 0.988 | 0.130 | 0.167 | 0.159 | 0.179 |
| Adv. Deb. | 0.670 | 0.854 | 0.818 | 0.219 | 0.246 | 0.108 | 0.130 |

both in group and individual fairness. We conclude that even in problems where neural networks outperform GBDT, BuDRO is an effective method when taking fairness into consideration.

## 6 SUMMARY AND DISCUSSION

We developed a gradient boosting algorithm that enforces individual fairness. The main challenge of enforcing individual fairness is searching for differential performance in the ML model, and we overcome the non-smoothness of the ML model by restricting the search space to a finite set. Unlike most methods for enforcing individual fairness, our method accepts non-smooth ML models. We note that the restricted adversarial cost function developed for fair gradient boosting may be used to audit other non-smooth ML models (*e.g.* random forests) for differential performance, but we defer such developments to future work. Theoretically, we show that the resulting fair gradient boosting algorithm converges globally and generalizes. Empirically, we show that the method preserves the accuracy of gradient boosting while improving group and individual fairness metrics.

## ACKNOWLEDGEMENTS

This paper is based upon work supported by the National Science Foundation (NSF) under grants no. 1830247 and 1916271. Any opinions, findings, and conclusions or recommendations expressed in this paper are those of the authors and do not necessarily reflect the views of the NSF.

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

# A    PROOFS OF THEORETICAL RESULTS

## A.1    NUMERICAL CONVERGENCE RESULTS

In this section, we focus on analyzing the numerical convergence of our boosting algorithm. To do so, we will work in the framework similar to that found in previous boosting literature (e.g. Mason et al. (1999)). To be concrete, we assume that $\mathcal{Y} \subset \mathbb{R}$. Then, we consider all base classifiers to be of the form $\mathbf{h} \colon \{x_1, \ldots, x_n\} \to \mathcal{Y}$ and thus treat a base classifier $\mathbf{h}$ as a vector $\mathbf{h} \in \mathbb{R}^n$ (precisely, $\mathbf{h} = [h(x_1), \ldots, h(x_n)]^T$). Moreover, we define the vector $\mathbf{y} = [y_1, \ldots, y_n]^T \in \mathbb{R}^n$. In this framework, the loss $\ell$ takes the form $\bar{\ell} \colon \mathbb{R}^n \times \mathbb{R}^n \to \mathbb{R}$, defined by $\bar{\ell}(\mathbf{f}, \mathbf{y}) = \frac{1}{n} \sum_{i=1}^{n} \ell(f_i, y_i)$. We abuse notation and write $L(\mathbf{f})$ for the robust loss function; this is a functional on $\mathbb{R}^n$ since it only depends on the values $f(x_i)$ for $x_i \in \{x_1, \ldots, x_n\}$.

In this context, given a functional $F \colon \mathbb{R}^n \to \mathbb{R}$, the functional derivative $\nabla F$ is the gradient of $F$. Additionally, $\|\cdot\|$ denotes the standard Euclidean norm on $\mathbb{R}^n$.

We require several assumptions for these results:

**Assumption A.1.** *(i) $\bar{\ell}$ is convex as well as first- and second-order differentiable with respect to $f$.*
*(ii) For any $\mathbf{v} \in \mathbb{R}^n$, $\bar{\ell}$ is $\omega_1$-Lipschitz differentiable, meaning there exists a constant $\omega_1 > 0$ such that for any $\mathbf{f}, \bar{\mathbf{f}} \in \mathbb{R}^n$*

$$\|\nabla \bar{\ell}(\mathbf{f}, \mathbf{v}) - \nabla \bar{\ell}(\bar{\mathbf{f}}, \mathbf{v})\| \le \omega_1 \|\mathbf{f} - \bar{\mathbf{f}}\|.$$

The assumption that the loss is convex is restrictive, and we only invoke it here to show global convergence. In section 3, we also assume the loss function is bounded. This pair of assumptions (convexity and boundedness) is not particularly restrictive because we have in mind a bounded input space (see Assumption 3.1).

We additionally require the following assumption, which is standard in the gradient descent literature (Nocedal & Wright, 2006). Essentially, we suppose that our boosting algorithm moves in a sufficient descent direction at each iteration. That is, the weak learner can ensure the angle between the searching direction and the functional gradient is bounded.

**Assumption A.2.** *There is a value $\delta > 0$, such that Algorithm 1 can find a function $\mathbf{h}_t$ with*

$$\theta_t := -\frac{\langle \nabla L(\mathbf{f}_t), \mathbf{h}_t \rangle}{\|\nabla L(\mathbf{f}_t)\| \cdot \|\mathbf{h}_t\|} \ge \delta, \quad \text{for all } t = 1, \cdots, T. \tag{A.1}$$

The following convergence result is a consequence of general results on the convergence of functional gradient descent algorithm for minimizing cost functionals (Mason et al., 1999). It states that, with a specific step-size, Algorithm 1 will converge to some classifier $f^*$.

**Theorem A.3.** *Under Assumptions A.1 and A.2, suppose we run Algorithm 1 with step-sizes*

$$\alpha_t = -\frac{\langle \nabla L(\mathbf{f}_t), \mathbf{h}_t \rangle}{\omega_1 \|\mathbf{h}_t\|^2}. \tag{A.2}$$

*Then, the optimization converges, i.e.:* $\lim_{t \to \infty} \nabla L(\mathbf{f}_t) = 0$.

This framework also results in the following, which shows that we obtain a global optimal solution.

**Theorem A.4.** *Under Assumptions A.1 and A.2, suppose $\{\mathbf{f}_t\}$ is the sequence generated by Algorithm 1 with step-size given by (A.2). Then, any stationary point $\{\mathbf{f}_t\}$ is the global optimal solution.*

The specific step-size presented above is only required for analyzing the convergence properties of Algorithm 1, which can be easily replaced by a step-size selected by some line search methods.

We begin the proof of numerical results by providing following preparatory properties of the cost functional $L$.

**Lemma A.5.** *Under Assumption A.1, it holds*

*(i) $L$ is convex, first- and second-order differentiable with respect to $\mathbf{f}$.*

*(ii)* $L$ *is a $\omega_1$-Lipschitz differentiable functional, that is*

$$\|\nabla L(\mathbf{f}) - \nabla L(\bar{\mathbf{f}})\| \leq \omega_1 \|\mathbf{f} - \bar{\mathbf{f}}\| \tag{A.3}$$

*for any $\mathbf{f}, \bar{\mathbf{f}} \in \mathcal{F}$.*

*Proof of Lemma A.5.* For part (i), recall $L(\mathbf{f}) = P^T \bar{\ell}(\mathbf{f}, \mathbf{y})$, combining with Assumption A.1 (i), we have

$$\nabla L(\mathbf{f}) = \text{diag}(P)\nabla\ell(\mathbf{f}) \quad \text{and} \quad \nabla^2 L(\mathbf{f}) = \text{diag}(P)\nabla^2\bar{\ell}(\mathbf{f}, \mathbf{y}),$$

where $\text{diag}(P)$ is a diagonal matrix with $[\text{diag}(P)]_{ii} = P_i$ for $i = 1, \cdots, n$. Since $P \in [0,1]^n$ and $\sum_{i=1}^{n} P_i = 1$, $L(\mathbf{f})$ is a linear combination of $\ell(f_i, y_i)$. Therefore, $L$ is a convex functional of $\mathbf{f}$.

For part (ii), according to Assumption A.1(ii), $\bar{\ell}(\mathbf{f}, \mathbf{y})$ is $\omega_1$-Lipschitz differentiable with respect to $\mathbf{f}$, yielding

$$\|\nabla^2 \bar{\ell}(\mathbf{f}, \mathbf{y})\| \leq \omega_1. \tag{A.4}$$

Given any $\mathbf{f} \in \mathcal{F}$, it holds

$$\begin{aligned}
\|\nabla^2 L(\mathbf{f})\| &= \|\text{diag}(P)\nabla^2\bar{\ell}(\mathbf{f}, \mathbf{y})\| \\
&\leq \|\text{diag}(P)\| \cdot \|\nabla^2\bar{\ell}(\mathbf{f}, \mathbf{y})\| \\
&\leq \omega_1,
\end{aligned} \tag{A.5}$$

where the first inequality follows from Cauchy–Schwarz inequality and the second inequality follows from Assumption (A.4) and $P \in [0,1]^n$. Combining (A.5) with the mean value theorem (for functionals) (Jerri, 1999), it holds

$$\begin{aligned}
\|\nabla L(\mathbf{f}) - \nabla L(\bar{\mathbf{f}})\| &\leq \|\nabla^2 L\left(c\mathbf{f} + (1-c)\bar{\mathbf{f}}\right)\| \cdot \|\mathbf{f} - \bar{\mathbf{f}}\| \\
&\leq \omega_1 \|\mathbf{f} - \bar{\mathbf{f}}\|,
\end{aligned}$$

where $c \in [0,1]$. It completes the proof. $\square$

*Proof of Theorem A.3.* By Lemma A.5, we have following lower bound for the difference between two successive loss functions:

$$\begin{aligned}
L(\mathbf{f}_t) - L(\mathbf{f}_{t+1}) &\geq L(\mathbf{f}_t) - \left( L(\mathbf{f}_t) + \alpha_t \langle \nabla L(\mathbf{f}_t), \mathbf{h}_t \rangle + \frac{\omega_1(\alpha_t)^2}{2}\|\mathbf{h}_t\|^2 \right) \\
&= -\alpha_t \langle \nabla L(\mathbf{f}_t), \mathbf{h}_t \rangle - \frac{\omega_1(\alpha_t)^2}{2}\|\mathbf{h}_t\|^2.
\end{aligned}$$

We choose $\alpha_t = -\frac{\langle \nabla L(\mathbf{f}_t, \mathbf{h}_t \rangle}{\omega_1 \|\mathbf{h}_t\|^2}$ as (A.2) to make the greatest reduction, which yields

$$L(\mathbf{f}_t) - L(\mathbf{f}_{t+1}) \geq \frac{\langle \nabla L(\mathbf{f}_t), \mathbf{h}_t \rangle^2}{2\omega_1 \|\mathbf{h}_t\|^2}. \tag{A.6}$$

Combining (A.6) with the sufficient decrease condition (A.1), we have

$$L(\mathbf{f}_t) - L(\mathbf{f}_{t+1}) \geq \frac{\delta^2 \|\nabla L(\mathbf{f}_t)\|^2}{2\omega_1}. \tag{A.7}$$

Summing up both sides of (A.7) from 0 to $\infty$, we have

$$\sum_{t=0}^{\infty} \|\nabla L(\mathbf{f}_t)\|^2 \leq \sum_{t=0}^{\infty} \frac{2\omega_1}{\delta^2}(L(\mathbf{f}_t) - L(\mathbf{f}_{t+1})) \leq \frac{2\omega_1}{\delta^2}L(\mathbf{f}_0) < \infty,$$

where the second inequality follows by $\mathbf{f}_t \geq 0$ for all $t$, yielding

$$\lim_{t\to\infty} \nabla L(\mathbf{f}_t) = 0.$$

$\square$

*Proof of Theorem A.4.* Suppose $\mathbf{f}_*$ is a stationary point of $\{\mathbf{f}_t\}$. We prove this theorem by contradiction by assuming that we can find a point $\hat{\mathbf{f}} \in \mathrm{lin}(\mathcal{F})$ such that $L(\hat{\mathbf{f}}) < L(\mathbf{f}_*)$.

The directional derivative of $L$ at $\mathbf{f}_*$ in the direction $\hat{\mathbf{f}} - \mathbf{f}_*$ is given by

$$\langle \nabla L(\mathbf{f}_*), \hat{\mathbf{f}} - \mathbf{f}_* \rangle = \lim_{\zeta \downarrow 0} \frac{L(\mathbf{f}_* + \zeta(\hat{\mathbf{f}} - \mathbf{f}_*)) - L(\mathbf{f}_*)}{\zeta}.$$

Since $L$ is convex, it holds

$$\langle \nabla L(\mathbf{f}_*), \hat{\mathbf{f}} - \mathbf{f}_* \rangle \leq \lim_{\zeta \downarrow 0} \frac{\zeta L(\hat{\mathbf{f}}) + (1 - \zeta) L(\mathbf{f}_*) - L(\mathbf{f}_*)}{\zeta}$$
$$= L(\hat{\mathbf{f}}) - L(\mathbf{f}_*) < 0.$$

Therefore, $\nabla L(\mathbf{f}_*) \neq 0$, which makes the contradiction. $\qquad\square$

## A.2 PROOFS OF GENERALIZATION ERROR BOUNDS

*Proof of Theorem 3.4.* Under Assumptions 3.1, 3.2, 3.3, we have the following result from Proposition 3.2 in Yurochkin et al. (2020) (recall Equations (2.2) and (2.4))

$$|L_e(f) - L_r(f)| \to 0 \quad \text{as} \quad n \to \infty.$$

Then, our proof is completed by showing $L$ converges to $L_e$ uniformly in $\mathcal{F}$. It has been shown (Blanchet & Murthy, 2016) that the dual to the optimization $L_e$ defined in (2.4) is given by

$$L_f(h) = \inf_{\lambda \geq 0} \lambda \epsilon - \frac{1}{n} \sum_{i=1}^{n} \sup_{x \in \mathcal{X}} \ell(h(x), y_i) + \lambda d_x^2(x, x_i). \tag{A.8}$$

Likewise, calculations (see Appendix B.3) reveal that the dual to (2.5) is given by

$$L(h) = \inf_{\lambda \geq 0} \lambda \epsilon - \frac{1}{n} \sum_{i=1}^{n} \max_{x \in \{x_1, \ldots, x_n\}} \ell(h(x), y_i) + \lambda d_x^2(x, x_i). \tag{A.9}$$

We thus need to establish a bound on

$$\delta_n = \left| \inf_{\lambda \geq 0} \lambda \epsilon - \frac{1}{n} \sum_{i=1}^{n} \sup_{x \in \mathcal{X}} \ell(h(x), y_i) + \lambda d_x^2(x, x_i) - \right.$$
$$\left. \inf_{\lambda \geq 0} \left[ \lambda \epsilon - \frac{1}{n} \sum_{i=1}^{n} \max_{x \in \{x_1, \ldots, x_n\}} \ell(h(x), y_i) + \lambda d_x^2(x, x_i) \right] \right|. \tag{A.10}$$

To do so, let $\lambda_n$ be a minimizer of (A.9). Then, we have that

$$\delta_n \leq \left| \lambda_n \epsilon - \frac{1}{n} \sum_{i=1}^{n} \sup_{x \in \mathcal{X}} \ell(h(x), y_i) + \lambda_n d_x^2(x, x_i) - \right.$$
$$\left. \lambda_n \epsilon + \frac{1}{n} \sum_{i=1}^{n} \max_{x \in \{x_1, \ldots, x_n\}} \ell(h(x), y_i) + \lambda_n d_x^2(x, x_i) \right|$$
$$= \left| \frac{1}{n} \sum_{i=1}^{n} \max_{x \in \{x_1, \ldots, x_n\}} \ell(h(x), y_i) + \lambda_n d_x^2(x, x_i) - \sup_{x \in \mathcal{X}} \ell(h(x), y_i) + \lambda_n d_x^2(x, x_i) \right| \tag{A.11}$$

Define a map $T$ on $\mathcal{D}$ such that $T(x_i) \in \arg\sup_{x \in \mathcal{X}} \ell(h(x), y_i) + \lambda_n d_x^2(x, x_i)$. By assumption, we have that

$$P_*(B_{n^{-1/2d}}(T(x_i)) \times \mathcal{Y}) \geq \delta/\sqrt{n}. \tag{A.12}$$

Thus, the probability of the event $\{\{x_1, \ldots, x_n\} \cap \bigcup_i B_{n^{-1/2d}}(T(x_i)) = \emptyset\}$ (*i.e.* the event that there are no points in the training data in $B_{n^{-1/2d}}(T(x_i))$) is at most $\sum_{i=1}^{n} (1 - \delta/n^{1/2})^n = n(1 - \delta/n^{1/2})^n$.

Thus assume that, for each $i$, there is a point $(x_i^*, y_i^*) \in \mathcal{D}$ such that $x_i^* \in B_{n^{-1/2d}}(T(x_i))$. Then, continuing from (A.11), we have that

$$\delta_n \leq \frac{1}{n} \sum_{i=1}^{n} |\ell(h(x_i^*), y_i) - \ell(h(T(x_i)), y_i)| + \left| \lambda_n(d_x^2(x_i^*, x_i) - d_x^2(T(x_i), x_i)) \right| \tag{A.13}$$

$$\leq \frac{1}{n} \sum_{i=1}^{n} \omega_2 d_x(x_i^*, T(x_i) + |\lambda_n(d(x_i^*, x_i) - d(T(x_i), x_i))(d(x_i^*, x_i) + d(T(x_i), x_i))| \tag{A.14}$$

$$\leq \frac{1}{n} \sum_{i=1}^{n} \frac{\omega_2}{n^{1/2d}} + \left| \frac{2\lambda_n \mathrm{diam}(\mathcal{X})}{n^{1/2d}} \right|. \tag{A.15}$$

$$\leq \frac{\omega_2}{n^{1/2d}} + \frac{2\omega_2 \mathrm{diam}(\mathcal{X})}{n^{1/2d}\sqrt{\epsilon}}. \tag{A.16}$$

In line (A.14), we use the fact that $\ell$ is $\omega_2$-Lipschitz with respect to $d_x$ (Assumption 3.2(ii)). In line (A.15), we use the fact that $x_i^* \in B_{n^{-1/2d}}(T(x_i))$ with the triangle inequality (to get that $d(x_i^*, x_i) - d(T(x_i), x_i) \leq d(x_i^*, T(x_i))$) and with the fact that $d(T(x_i), x_i) \leq \mathrm{diam}(\mathcal{X})$. Finally, in line (A.16), we apply Lemma A.1 from Yurochkin et al. (2020) which asserts that $0 \leq \lambda_n \leq \frac{\omega_2}{\sqrt{\epsilon}}$. This completes the proof of the desired result. $\square$

Compared to most theoretical studies of distributionally robust optimization, the proof of Theorem 3.4 is complicated by the restriction of the $c$-transform to the sample $\mathcal{D}$ in (A.9). This complication arises because the max over the sample is generally (much) smaller than the max over the sample space due to the curse of dimensionality. This discrepancy between the max over the sample and the max over the space space is responsible for the dependence of the rate of convergence on the dimension of the feature space in Theorem 3.4.

## B FURTHER IMPLEMENTATION CONSIDERATIONS

Here we present a method based on entropic regularization that can be used to quickly obtain an approximate solution to the linear program in Equation 2.8. Moreover, this method requires only simple matrix computations; thus, we are able to implement it in TensorFlow for quick GPU computations.

It is also possible to quickly find the solution to the dual of the linear program (2.8). Obtaining the primal optimizer $\Pi^*$ from the dual solution, however, can be difficult. For this reason, the Sinkhorn-based entropic regularization method runs significantly more quickly than the dual method in our implementations if we insist on using complementary slackness to determine the exact primal optimizer $\Pi^*$. In practice, we can quickly obtain an approximation to $\Pi^*$ from the dual solution, however (Algorithm 2). A discussion of the dual method can be found in Section B.3.

### B.1 ENTROPIC REGULARIZATION

Equation (2.8) is an optimization problem over probability distributions. Thus, it is reasonable to regularize the objective function in Equation (2.8) with the entropy of the distribution.

Formally, define the matrix $R$ by $R_{i,j} = \ell(f, (x_i, y_j))$ - this is the loss incurred if point $j$ with label $y_j$ is transported to point $i$. Moreover, define the matrix $C$ by $C_{ij} = d_x^2(x_i, x_j)$, and let $\gamma$ denote a regularization parameter. With this notation, including entropic regularization, the problem becomes:

$$\Pi^* = \arg \max_{\Pi \in \Gamma} \langle R, \Pi \rangle - \gamma \langle \log(\Pi), \Pi \rangle \tag{B.1}$$

where $\log(\Pi)_{ij} = \log(\Pi_{ij})$.

Adding the entropy of $\Pi$ to the objective encourages the optimizer $\Pi^*$ to be less sparse than the (low entropy) optimizer of the original optimal transport problem. Note that it is not inherently desirable to find a sparse optimizer $\Pi^*$, since we only consider the marginals of $\Pi^*$ while boosting. Coming close to maximizing the original objective $\langle R, \Pi^* \rangle$ is far more important than sparsity.

We follow the Sinkhorn method to develop a solution to the problem (B.1). The Lagrangian is given by

$$\mathcal{L}(\Pi, \lambda, \eta) = \sum_{ij} R_{ij}\Pi_{ij} - \gamma \sum_{ij} \log(\Pi_{ij})\Pi_{ij} - \sum_j \lambda_j(\sum_i \Pi_{ij} - \tfrac{1}{n}) - \eta(\sum_{ij} C_{ij}\Pi_{ij} - \epsilon) \quad \text{(B.2)}$$

so that at the optimum we have

$$\frac{\partial \mathcal{L}}{\partial \Pi_{ij}} = R_{ij} - \gamma - \gamma \log(\Pi_{ij}) - \lambda_j - \eta C_{ij} = 0 \quad \text{(B.3)}$$

which means that

$$\Pi_{ij}^* = \exp(R_{ij}/\gamma)\exp(-\lambda_j/\gamma - 1)\exp(-\eta C_{ij}/\gamma). \quad \text{(B.4)}$$

Now, define matrices $V$ and $K$ and a vector $u$ by the following expressions:

- $V_{ij} = \exp(R_{ij}/\gamma)$,
- $K_{ij} = \exp(-\eta C_{ij}/\gamma)$, and
- $u_j = \exp(-\lambda_j/\gamma - 1)$.

By definition, we have that $\Pi_{ij}^* = V_{ij}K_{ij}u_j$, and that $V$ is a constant (it does not depend on any of the variables of the Lagrangian). Exploiting the constraints of the set $\Gamma$ (see the text before (2.8)), we see that $u$ can be written in terms of $K$:

$$\sum_i \Pi_{ij} = u_j \sum_i K_{ij}V_{ij} = \tfrac{1}{n} \Rightarrow \quad u_j = \frac{1}{n\sum_i K_{ij}V_{ij}}. \quad \text{(B.5)}$$

Thus, the solution $\Pi^*$ depends only on $K$. Since $K$ is determined completely by the Lagrange multiplier $\eta$, we need only to find the value of $\eta$ that makes the other constraint of the set $\Gamma$ tight:

$$\sum_{ij} C_{ij}\Pi_{ij} = \sum_{ij} C_{ij}V_{ij}K_{ij}u_j = \mathbf{1}_n^T(C \odot K(\eta) \odot V)u = \epsilon. \quad \text{(B.6)}$$

where $A \odot B$ denotes entrywise (Hadamard) product of the matrices $A$ and $B$. We use a root finding algorithm to determine the optimal value of $\eta$: specifically, we find the root of $m(\eta) = \epsilon - \mathbf{1}_n^T(C \odot K(\eta) \odot V)u$. In our experiments, the bisection or secant methods generally converge in only around 10 to 20 evaluations of $m(\eta)$.

A fast method for evaluating $m(\eta)$ is presented in Algorithm 4. It consists of simple entrywise matrix operations (entrywise multiplications and exponentiations); this is highly amenable to processing on a GPU, and we have implemented this Sinkhorn-based method with TensorFlow. In Algorithm 5, we provide a method for using the optimal root $\eta^*$ to obtain the coupling matrix $\Pi$ following Equation (B.4).

---

**Algorithm 4** Fast evaluation of Sinkhorn objective

---

1: **Input**: $\eta \geq 0$; cost matrix $C$; loss matrix $R$; tolerance $\epsilon$; regularization strength $\gamma$.
2: Let $T \leftarrow \exp(R/\gamma - \eta C/\gamma)$ {entrywise exponentiation}
3: Let $u \leftarrow 1/(n\mathbf{1}_n^T \cdot T)$ {entrywise reciprocation}
4: **return** $\epsilon - \mathbf{1}_n^T \cdot (C \odot T) \cdot u$

---

**Algorithm 5** Find $\Pi^*$ using entropic regularization via (B.4)

---

1: **Input**: cost matrix $C$; loss matrix $R$; tolerance $\epsilon$; regularization strength $\gamma$.
2: Let $\eta^*$ be the root of Algorithm 4 with fixed inputs $C$, $R$, $\epsilon$, and $\gamma$.
3: Let $S \leftarrow \exp(R/\gamma - \eta^* C/\gamma)$.
4: Let $u \leftarrow 1/(n\mathbf{1}_n \cdot S)$.
5: Define $\Pi$ by $\Pi_{ij} = S_{ij}u_j$.
6: **return** $\Pi$

---

### B.2 STOCHASTIC GRADIENT DESCENT FOR FINDING $\eta^*$ WITH ENTROPIC REGULARIZATION

Although the Sinkhorn-based method presented above is fast, note that it is still not especially scalable, as it requires holding at least two $n \times n$ matrices ($R$ and $C$) in memory at the same time. Assuming double precision floating point arithmetic, each of these matrices requires more than 10 GB of memory when $n$ is approximately $4 \times 10^4$.

Thus, following Genevay et al. (2016), we express the determination of the optimal dual variable $\eta^*$ as the minimization of an expectation over the empirical distribution $P_n$; thus, it is possible to leverage stochastic gradient descent (with mini-batching) to quickly find the optimal dual variable $\eta^*$ while using a small amount of memory.

To develop this expression, note that the Lagrangian dual to the problem (B.1) is given by

$$\min_{\eta \geq 0, \lambda} \left[ \max_{\Pi} \mathcal{L}(\Pi, \lambda, \eta) \right] \tag{B.7}$$

where $\mathcal{L}$ is defined in Equation (B.2). The optimal value of $\Pi_{ij}^*$ for the inner maximum is established in Equation (B.4). Using this optimal value, we see that

$$\log(\Pi_{ij}^*) = \frac{1}{\gamma} \left( R_{ij} - \lambda_j - \eta C_{ij} \right) - 1 \tag{B.8}$$

and we can use this to calculate that

$$\mathcal{L}(\Pi^*, \lambda, \eta) = \eta\epsilon + \frac{1}{n} \sum_j \lambda_j + \gamma \sum_{ij} \Pi_{ij}^* = \eta\epsilon + \frac{1}{n} \sum_j \lambda_j + \gamma \sum_{ij} \exp\left( \frac{R_{ij} - \lambda_j - \eta C_{ij}}{\gamma} - 1 \right). \tag{B.9}$$

To minimize this with respect to $\lambda$ (which is unconstrained) we set

$$\frac{\partial}{\partial \lambda_j} (\mathcal{L}(\Pi^*, \lambda, \eta)) = \frac{1}{n} - \exp\left( \frac{-\lambda_j}{\gamma} \right) \sum_i \exp\left( \frac{R_{ij} - \eta C_{ij}}{\gamma} - 1 \right) = 0 \tag{B.10}$$

which means that

$$\exp\left( \frac{-\lambda_j^*}{\gamma} \right) = \frac{1}{n} \cdot \left( \sum_i \exp\left( \frac{R_{ij} - \eta C_{ij}}{\gamma} - 1 \right) \right)^{-1}$$

$$\Rightarrow \quad \lambda_j^* = -\gamma \log \frac{1}{n} + \gamma \log \left( \sum_i \exp\left( \frac{R_{ij} - \eta C_{ij}}{\gamma} - 1 \right) \right).$$

Thus, substituting into (B.9), we calculate that

$$\mathcal{L}(\Pi^*, \lambda^*, \eta) = \gamma + \sum_j \frac{1}{n} \left( \eta\epsilon + \gamma \log \left( \sum_i \exp\left( \frac{R_{ij} - \eta C_{ij}}{\gamma} - 1 \right) \right) - \gamma \log \frac{1}{n} \right) \tag{B.11}$$

The value $\eta \geq 0$ that minimizes (B.11) is the same as the minimizer of

$$\min_{\eta \geq 0} \mathbb{E}_{(x,y) \sim P_n} \left[ \eta\epsilon + \gamma \log \left( \sum_i \exp\left( \frac{\ell(f, (x_i, y)) - \eta d_x^2(x_i, x)}{\gamma} \right) \right) \right] \tag{B.12}$$

where we ignored some constants that don't affect the minimizing value $\eta^*$ and substituted in the definitions of $R_{ij}$ and $C_{ij}$ (see the paragraph at the start of Section B.1).

The problem B.12 is amenable to minimization via stochastic gradient descent (SGD) - this is presented in Algorithm 6. In every descent step of Algorithm 6, we are only working with a subset of the columns of $R$ and $C$. Thus, $R$ and $C$ may be stored anywhere (or even computed on-the-fly) - they do not need to be kept in RAM or sent to the GPU memory. Thus, this SGD version can be used for essentially arbitrarily large data sets. Empirically, we also observe good results from running only a few gradient descent steps to obtain an approximation to $\eta^*$ in every boosting step; this allows the SGD method to run as quickly as (or more quickly than) the normal Sinkhorn method (Algorithm 4). To find the final transport map $\Pi$, we use Algorithm 5, as before.

---

**Algorithm 6** SGD to find optimal dual variable with entropic regularization $\eta^*$

---

1: **Input**: Starting point $\eta_1 > 0$; cost matrix $C$; loss matrix $R$; tolerance $\epsilon$; regularization strength $\gamma$; batch size $B$, step sizes $\alpha_t > 0$.
2: **repeat**
3:   Sample indices $j_1, \ldots, j_B$ uniformly at random from $\{1, \ldots n\}$.
4:   Let $R_t \leftarrow$ columns $j_1, \ldots, j_B$ of $R$. Let $C_t \leftarrow$ columns $j_1, \ldots, j_B$ of $C$. $\{R_t, C_t \in \mathbb{R}^{n \times B}\}$
5:   Let $w_j^t(\eta) \leftarrow$ sum all elements in column $j$ of $\exp\left(\frac{1}{\gamma}(R_t - \eta C_t)\right)$ {entrywise exponentiation}
6:   $\eta_{t+1} \leftarrow \max\{0, \eta_t - \alpha_t \epsilon - \alpha_t \gamma \frac{d}{d\eta}\left[\sum_{j=1}^{B} \log w_j^t(\eta)\right]_{\eta=\eta_t}\}$
7: **until** converged

---

### B.3 Dual of robust empirical loss function $L$

Although the algorithms presented in the preceding section run quickly and can handle arbitrarily large inputs, they only obtain an approximation to the true optimal transport map (due to the entropic regularization). Here, we present a method to quickly obtain the solution to the dual of the original linear program (2.8). We find that constructing the transport map $\Pi^*$ from the knowledge of the dual optimum is difficult; however, there are quick methods to produce reasonable approximations to $\Pi^*$ from the knowledge of the dual solution.

Following standard methods, the dual of the linear program (2.8) is given by:

$$\inf_{\eta \geq 0} \epsilon\eta + \frac{1}{n} \sum_{j=1}^{n} \nu_j \tag{B.13}$$
$$\text{s.t. } \nu_j \geq R_{ij} - \eta C_{ij} \text{ for all } i, j$$

Which can be simplified to:

$$\inf_{\eta \geq 0} \epsilon\eta + \frac{1}{n} \sum_{j=1}^{n} \max_i R_{ij} - \eta C_{ij}. \tag{B.14}$$

Define $\lambda_j(\eta) = \max_i R_{ij} - \eta C_{ij}$ and let $M(\eta) = \epsilon\eta + \frac{1}{n} \sum_{j=1}^{n} \lambda_j(\eta)$ denote the dual objective function from (B.14). Note that each $\lambda_j(\eta)$ is a monotone decreasing convex piecewise linear function of $\eta$. In addition, since $C_{jj} = 0$, we have that $\lim_{\eta \to \infty} \lambda_j(\eta) \geq R_{jj}$; that is, the $\lambda_j$ are positive and eventually become constant.

This means that the infimum (over $\eta$) will either occur at $\eta = 0$ or at one of the *corners* of one of the $\lambda_j$ (a *corner* is a value of $\eta$ such that there exist $i_1$ and $i_2$, $i_1 \neq i_2$, with $R_{i_1 j} - \eta C_{i_1 j} = R_{i_2 j} - \eta C_{i_2 j} = \max_i R_{ij} - \eta C_{ij}$; i.e. at least two of the lines that define $\lambda_j$ are intersecting and creating a point where $M(\eta)$ is not differentiable). Thus, it is theoretically possible to exactly obtain the optimizer $\eta^*$ by enumerating and testing all of these corners.

In practice, we have found that it is quicker to approximate $\eta^*$ by noticing that an element of the subgradient for $M(\eta)$ is given by

$$\epsilon - \frac{1}{n} \sum_j C_{i_j, j} \quad : \quad i_j \in \arg\max_i R_{ij} - \eta C_{ij}. \tag{B.15}$$

Thus, a bisection method can be implemented to approximate the point $\eta^*$ such that $\frac{d}{d\eta} M(\eta) < 0$ for $\eta < \eta^*$ and $\frac{d}{d\eta} M(\eta) > 0$ for $\eta > \eta^*$. This is the corner that we are looking for. Using a bisection method (or similar) guarantees that we only need a small fixed number ($\approx 30$) of subgradient computations in every boosting step.

Complementary slackness can directly be exploited to quickly find the desired solution $\Pi^*$ to the original discrete SenSR LP from the dual optimizer $\eta^*$ if the following conditions hold:

1. There is an index $j$ such that the point $\eta^* = \arg\min_{\eta \geq 0} M(\eta)$ is a corner of $\lambda_j$ with $|\arg\max_i R_{ij} - \eta^* C_{ij}| = 2$ (only two of the lines that define $\lambda_j$ intersect at $\eta_*$).

2. For all $k \neq j$, the point $\eta^*$ is not a corner of $\lambda_k$ (i.e. $|\arg\max_i R_{ik} - \eta^* C_{ik}| = 1$).

These conditions are based on the fact that, for all $i$ and $j$, $\Pi_{ij}$ is the primal variable corresponding to the constraint $v_j \geq R_{ij} - \eta C_{ij}$ in the dual (B.13). Condition 2 implies that, for all $k \neq j$, there is a value of $t$ such that $\Pi_{tk}^* = \frac{1}{n}$ and $\Pi_{ik}^* = 0$ for all $i \neq t$. Then, condition 1 implies that there are only two nonzero values in column $j$ of $\Pi^*$. These two nonzero values must sum to $\frac{1}{n}$ and the constraint $\langle C, \Pi^* \rangle = \epsilon$ must be satisfied. With the complete knowledge of the other values of $\Pi^*$, this results in two linear equations with two unknowns (the two remaining nonzero values of $\Pi^*$), and this system can be easily solved to give the exact solution $\Pi^*$.

Similar tricks can be applied to rapidly solve for $\Pi^*$ when slightly more entries are candidates for being nonzero (according to complementary slackness). Unfortunately, since $d_x$ is a fair distance (and thus it usually ignores several directions in $\mathcal{X}$), it is often the case that there are multiple individuals $x_i \neq x_j$ in the training data such that $d_x(x_i, x_j) = 0$ (i.e. there are a significant number of off-diagonal entries of $C$ that are 0). In practice, we have observed that these points also often produce the same predicted value $f(x_i) = f(x_j)$ after the first step of boosting. This results in a situation where complementary slackness presents a complicated system of equations for obtaining the primal optimizer $\Pi^*$.

Thus, in order to maintain an efficient boosting algorithm in practice, we use the dual optimizer $\eta^*$ to construct an approximation $\hat{\Pi}$ to the primal optimizer $\Pi^*$ using the following heuristic: for all $j$, randomly select $t$ in $\arg\max_i R_{ij} - \eta^* C_{ij}$ and let $\hat{\Pi}_{tj} = \frac{1}{n}$. That is, in each column, randomly select one of the candidate nonzero entries (according to complementary slackness) and set it to $\frac{1}{n}$ in $\hat{\Pi}$. This approximation $\hat{\Pi}$ actually is an interpretable transport map: each point in the training data is mapped (completely) to another point in the training data. We present the construction of $\hat{\Pi}$ using Algorithm 2 in the main text, and we use Algorithm 2 in our experiments on the German credit data set the Adult data set in Section 5. An outline of the full dual method (not using SGD) is presented in Algorithm 7. Although it is not exactly clear as to how good (or bad) of an approximation $\hat{\Pi}$ is to $\Pi^*$, our experimental results show that it is functional.

---

**Algorithm 7** Find $\Pi$ using the dual formulation (following (B.15))

1: **Input**: cost matrix $C$; loss matrix $R$; tolerance $\epsilon$; root tolerance $\delta$
2: Use the bisection method (or something similar) to find a value $\eta^*$ such that for all $\eta$
  - $\sup\{\epsilon - \frac{1}{n}\sum_j C_{i_j j} : i_j \in \arg\max_i R_{ij} - \eta C_{ij}\} < 0$ when $\eta < \eta^* - \delta$
  - $\inf\{\epsilon - \frac{1}{n}\sum_j C_{i_j j} : i_j \in \arg\max_i R_{ij} - \eta C_{ij}\} > 0$ when $\eta > \eta^* + \delta$.
3: Obtain $\Pi$ using steps 8-13 of Algorithm 2.
4: **return** $\Pi$

---

## C  EXPERIMENTAL DETAILS

In this section, we provide the details of the experiments using the BuDRO algorithm. We start with a synthetic motivation, then discuss the three data sets that are presented in the main text.

### C.1  SYNTHETIC MOTIVATION

Consider a data set with two features, $x_1$ and $x_2$. Suppose that the one feature $x_1$ is protected and the other feature $x_2$ is can be used for making fair decisions. For example: our task may be to decide which individuals to approve for a loan. The protected feature $x_1$ may correspond to the percentage of nonwhite residents in the applicant's zip code, and the other feature $x_2$ may correspond to the applicant's credit score (or some other fair measure of credit worthiness). In this case, we can approximate a fair metric $d_x$ by the difference in the second feature: the fair distance between the two individuals $(x_1^a, x_2^a)$ and $(x_1^b, x_2^b)$ is given by $d_x((x_1^a, x_2^a), (x_1^b, x_2^b)) = |x_2^a - x_2^b|$.

We synthetically constructed such a data set in the following manner: 150 individuals were independently drawn from the same centered normal distribution. Let $R$ be a rectangle of minimum area containing the 150 points; let $L_1$ be the line passing through the top right corner and the bottom left

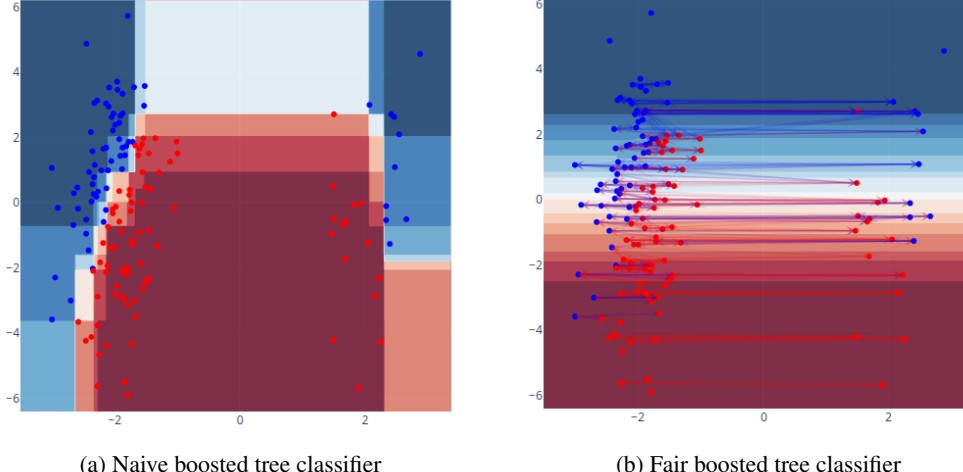

(a) Naive boosted tree classifier          (b) Fair boosted tree classifier

Figure 1: Comparison of GBDT classifiers without (a) and with (b) the fairness constraints introduced in this paper. The classifiers output a probability in $[0, 1]$ - these probabilities are discretized to binary labels to create a classification. In the figures, red points are individuals with true label 0 and blue points are individuals with true label 1. The darker red areas correspond to lower output probabilities and the darker blue areas correspond to higher output probabilities. The arrows in (b) indicate the transport map corresponding to the fairness constraint for the previous boosting step.

corner of $R$ and $L_2$ be the line passing through the top left corner and the bottom right corner of $R$. 125 of the samples were chosen (uniformly at random) to belong to majority white neighbourhoods: these individuals were labeled 0 if they were below $L_1$ and 1 if they were above $L_1$, and then they were all shifted to the left (by 2, so that the highly white cluster is centered at $(-2, 0)$). The remaining 25 individuals make up the majority nonwhite cluster: they were classified according to $L_2$ and then centered at $(2, 0)$.

In Figure 1, we show such an example of this setup. The red points in the figure correspond to individuals that are labeled 0; these represent individuals that should be declined for the loan (*e.g.* they have defaulted on a loan, with these data collected in the past to make future predictions). The blue points are individuals that are labeled 1 and represent people who should be approved (e.g. they have paid back a loan in the past). The horizontal axis represents the percentage of nonwhite residents in an individual's zip code, while the vertical axis is taken as some fair measure of credit worthiness.

The naively trained GBDT classifier in Figure 1(a) shows high accuracy on this synthetic data set, but it is unfair - requiring that individuals from neighbourhoods containing medium to high percentages of nonwhite individuals obtain a significantly higher credit score than those in very racially homogeneous neighbourhoods. Applying our individually fair gradient boosting algorithm to the data set, however, results in a fair classifier (visualized in Figure 1(b)). The credit score threshold for loan acceptance is consistent across the different racial make ups of zip codes. This provides a synthetic visualization of the performance of the BuDRO method, and suggests that it is working as we have described.

As a tangent: note that the clusters in this example are generated in a symmetric manner. This is a case where the unfairness in the naively trained classifier is coming from a lack of data and a push for extra accuracy rather than a specific inherent bias in the data.

## C.2 DETAILS COMMON TO ALL EXPERIMENTAL DATA SETS

**Fair metric** We follow one of the methods presented (Yurochkin et al., 2020) to construct fair metrics for the experimental data sets. Specifically, for a given data set, we determine a finite set $T$ of protected directions in $\mathcal{X}$. In a similar fashion to the synthetic example in C.1, changes in these protected directions should intuitively be ignored by a fair measure. Thus, the fair metric $d_x$ is defined by projecting onto the orthogonal complement of $\text{span}(T)$ and considering the Euclidean

distance on the projected space. In particular, let $A = \text{span}(T)$ and $\text{proj}(A)$ denote the projection onto $A$. Then,

$$d_x(x_1, x_2) = \|((I - \text{proj}(A)) \cdot (x_1 - x_2)\|_2. \tag{C.1}$$

The protected directions are determined in a similar fashion for each experimental data set. In particular, each data set contains (one or more) protected attributes: the indicator for each protected attribute is included in the set of protected directions[1].

We additionally obtain one or more extra protected directions in the following manner. For a fixed protected attribute $g$ (e.g. gender), we remove the feature $g$ from the data set and train a linear model on the edited data set to predict the removed feature $g$ (we use logistic regression with an $\ell_2$ regularization of strength 0.1 when $g$ is binary; ridge regression with cross validation is used for a non-binary feature $g$). Let $w$ be the normal vector to the separating hyperplane corresponding to this linear model: we include $w$ in $T$, the set of protected directions.

For example, on the Adult data set, we consider both the gender and race features as protected attributes (both features are binary, see the description in Section 5). The set $T$ for Adult then contains three protected directions. Two of these protected directions are given by $e_g$, the indicator vector of the gender feature, and $e_r$, the indicator vector of the race feature. We obtain the third protected direction $w$ by removing the gender feature from the data set and training logistic regression on this edited data set with targets given by the gender feature. Several of the features in the Adult data set (such as the is_husband and is_wife categories of the relationship feature) are highly predictive of gender. Combined with the gender imbalance in the data set, this allows for logistic regression to be able to predict gender with nearly 80% accuracy on a (holdout) test set. We thus take $w$ to be the normal vector to the separating hyperplane discovered during this logistic regression.

Note that a metric defined in this way will assign a distance of 0 between two individuals that differ only in the protected attributes (e.g. gender or race) but are identical in all other features. Additionally, if the difference $x_1 - x_2$ between two individuals is nearly parallel to one of the logistic regression directions $u$, then $d(x_1, x_2)$ will also be small. For example, on the Adult data set, the vector $w$ (defined in the previous paragraph) exhibits comparatively high support on the is_husband and is_wife categories of the relationship feature. Thus, if $x_1$ and $x_2$ are identical except that $x_1$ has wife = 1 and $x_2$ has husband = 1, then $d_x(x_1, x_2)$ will be small.

This fair metric $d_x$ is an approximation to an actual fair metric on $\mathcal{X}$: it does not capture information about all protected features, and only makes a heuristic approximation to reduce differences between true race and gender groups of individuals.

**Comparison methods**   Here we give an overview of the ML methods that we compare to BuDRO. Please refer to the included code for full information about the hyperparameter grids that we consider; see the respective data set sections for further information about training details and the optimal hyperparameter choices.

We consider three boosting methods in addition to BuDRO: baseline GBDTs, projecting, and reweiging. Our implementations of these methods all use the XGBoost framework (Chen & Guestrin, 2016). The GBDT hyperparameters that we consider are:

- max_depth, the maximum depth of the decision trees used as weak learners;
- lambda, an $\ell_2$ regularization parameter;
- min_weight, a tree regularization parameter; and
- eta, the XGBoost learning rate.

We additionally examine the effects of boosting for different numbers of steps. We always present the results of reweighing using the default XGBoost hyperparameters.

The *projecting* preprocessing method functions by eliminating the entire protected subspace from the data set before training a vanilla GBDT (see, for example Bower et al. (2018), Yurochkin et al. (2020), Prost et al. (2019)). In particular, using the notation from the discussion of fair metrics (above), we project all of the data onto the orthogonal complement of $\text{span}(T)$ as a preprocessing step. This has

---

[1]The indicator for an attribute is a vector with only one nonzero entry; that nonzero entry appears in the attribute that we are indicating.

the effect that the final classifier will be completely blind to differences in the protected directions in $T$. Our experiments (see e.g. Table 2) show that this is not enough to produce an individually fair classifier: changes to attributes that are highly correlated with elements of $T$ are still used to make classification decisions.

*Reweighing* is presented in Kamiran & Calders (2011). Essentially, this method functions by assigning weights to the individuals in the training data. These weights are chosen to force protected group status to appear statistically independent to the outcomes in $\mathcal{Y}$. Then a GBDT is trained on the reweighted data. This is inherently a group fairness method (the data are weighted to match a group fairness constraint) that cannot be used when the protected attribute does not take a finite number of values. We use the default XGBoost parameters along with reweighing to generate the values that are presented in the main text.

To test if GBDT classifiers are useful on the tabular data sets that we consider here, we also train (naive) one-layer (100 unit) fully connected neural networks on all data sets. For the Adult and the COMPAS data sets, we additionally compare to the SenSR method from Yurochkin et al. (2020) and the adversarial debiasing method from Zhang et al. (2018). The SenSR method creates an individually fair neural network through stochastic gradient descent on a robust loss similar to the one considered in this work. Adversarial debiasing is based on minimizing the ability to predict the protected attributes from knowledge of the final outputs of a predictor, and also draws on ideas from robustness in machine learning. It is constructed to provide improvements to statistical group fairness quantities rather than for the creation of an individually fair classifier.

**Evaluation metrics** We evaluate the ML methods with a type of individual fairness metric that we define below, also discussed Section 5 of the main text. It is not generally true that individual fairness will imply group fairness: it will depend on the group fairness constraint in question as well as the fair metric $d_x$ on $\mathcal{X}$. We thus also report several group fairness metrics to allow for comparison with other methods in the literature; it remains future work to establish the precise conditions under which individual fairness will imply group fairness.

The specific group fairness metrics that we consider here are $\text{GAP}_{\text{Max}}$ and $\text{GAP}_{\text{RMS}}$, as introduced in De-Arteaga et al. (2019). Suppose that there are two groups of individuals, labeled by $g = 0$ (a protected group) and $g = 1$ (a privileged group). Then, given true outcomes $Y$ and predicted outcomes $\hat{Y}$, we can consider a statistical fairness gap defined by

$$\text{Gap}_y = P(\hat{Y} = y | Y = y, g = 0)$$
$$- P(\hat{Y} = y | Y = y, g = 1)$$

for each possible outcome $y \in \mathcal{Y}$. Note that a large value of $|\text{Gap}_y|$ corresponds to (correctly) assigning the outcome $y$ to a larger fraction of individuals belonging one of the groups than the other. For example, suppose that $\mathcal{Y} = \{0, 1\}$, with an 1 indicating a favorable outcome. Then, a large value of $|\text{Gap}_1|$ means that our classifier is able to identify the successful individuals from one group at a higher rate than it is able to identify the successful individuals from the other. Thus, large values of $\text{Gap}_y$ indicate unfair performance by the classifier at the level of groups. We then define

$$\text{GAP}_{\text{Max}} = \max_{y \in \mathcal{Y}} |\text{Gap}_y|$$
$$\text{GAP}_{\text{RMS}} = \sqrt{\frac{1}{|\mathcal{Y}|} \sum_{y \in \mathcal{Y}} \text{Gap}_y^2}.$$

(C.2)

It is difficult to evaluate the individual fairness of an ML model since we only have access to an approximation to a true fair distance on $\mathcal{X}$ (we cannot, for example, consider pairs of training points that are definitely close to each other). We expect that the fair metrics considered in this work are good approximations to true fair metrics; nonetheless, we still desire to evaluate the individual fairness of an ML model in a way that is agnostic to the choice of approximate fair metric. For this reason, we construct counterfactual pairs of individuals (that is, pairs of individuals that differ only in certain protected attributes) from the input data. Intuitively, these counterfactual pairs should be treated the same way by any individually fair classifier (they should be 'close' according to the true fair metric).[2]

---

[2]This is somewhat debatable. For example, does flipping gender but keeping all other attributes the same truly result in an equivalent individual of a different gender?

We thus examine how often these counterfactual individuals are assigned to the same outcome. These evaluation criteria are inspired by Garg et al. (2018), which examined changes in predicted sentiment when changing one word in a sentence.

Specifically, suppose an attribute $g$ takes values in a set $V$. To measure the consistency of the predictor $f$ with respect to $g$ (the "$g$-consistency of $f$"), we construct $|V|$ copies of the test data. These copies are altered so that the value of the attribute $g$ is constant on each copy and so that each value in $V$ is represented in a copy of the data. We then apply $f$ to each copy of the data to obtain $|V|$ vectors of predicted outcomes $\hat{y}_1, \ldots, \hat{y}_{|V|}, \hat{y}_j \in \mathcal{Y}^n$. The $g$-consistency of $f$ is then the fraction of individuals who are assigned the same outcome in every copy of the data set. That is, it is the fraction of indices $i$ such that $(\hat{y}_1)_i = (\hat{y}_2)_i = \ldots = (\hat{y}_{|V|})_i$.

## C.3 GERMAN CREDIT

German credit is a data set that is commonly evaluated in the fairness literature (Dua & Graff, 2017). It contains information about 20 attributes (seven numerical, 13 categorical) from 1000 individuals; the ML task is to label the individuals as good or bad credit risks. Several of the features, such as `amount_in_savings` and `length_current_employment`, are numerical but have been recorded as categorical. Additionally, some of the other features, such as `employment_skill_level` and `credit_history_status` are categorical but could be considered to be ordered (for example, "all credits paid" is better than "past delays in payments" which is better than "account critical"). In some analyses, these ordered categorical features are converted to integers; for the purposes of this analysis, we one-hot encode all of the categorical features (which results in 62 features in our input data). We additionally standardize each numerical feature (that is, center by subtracting the mean and divide by the standard deviation). None of the data points are removed.

We treat `age` (a numerical feature that we standardize) as the protected attribute in the German credit data set. In order to report the group fairness metrics, we consider two protected groups: one group consists of individuals who are younger than 25, the other group consists of those who are 25 or older. This split was proposed by Kamiran & Calders (2009) to formalize the potential for age discrimination with this data set.

**Fair distance** For the German credit data set, we consider two protected directions, the first of which is the indicator vector $e_a$ of the `age` attribute. Additionally, following the framework described in C.2, we eliminate the `age` attribute from the data and use ridge regression to train a classifier to predict age from the other features (we use the default parameters in the `RidgeCV` class from the `scikit-learn` package, version `0.21.3` (Pedregosa et al., 2011)). The second protected direction $w$ is a normal vector to the hyperplane created by ridge regression (i.e. it is the vector of ridge regression coefficients).

**Evaluation metrics** As discussed above, we consider the $\mathrm{Gap}_{\mathrm{RMS}}$ and $\mathrm{Gap}_{\mathrm{MAX}}$ for a binarized version of the `age` attribute.

For an individual fairness metric, we cannot examine an age consistency, since age is a numerical feature. German credit contains a categorical `personal_status` feature that encodes some information about gender and marital status[3], however. After one-hot encoding, we find that several of the `personal_status` categories are well-correlated with the `age` attribute according to ridge regression. Specifically, the `male_divorced/separated` category is positively correlated with age (the ridge regression coefficient has an average of $0.26$ over our 10 train/test splits) and the `male_married/widowed` feature is negatively correlated with age (the ridge regression coefficient has an average of $-0.25$). Since this `personal_status` attribute can be considered to be a protected attribute, we examine a consistency measure based on this `personal_status` attribute as described in C.2. We refer to this consistency measure as *status consistency* (or *S-cons*).

**Method training and hyperparameter selection** The labels of the German credit data are quite unbalanced (only 30% of the labels are 1). We thus train all ML methods to optimize the balanced

---

[3]The `personal_status` categories are `male_single`, `male_married/widowed`, `male_divorced/separated`, `female_single`, and `female_divorced/separated/married`.

accuracy. For the GBDT methods (baseline, projecting, and BuDRO), this is accomplished by setting the XGBoost parameter `scale_pos_weight` to $\frac{0.7}{0.3}$, the ratio of zeros to ones in the labels of the training data. For naive (baseline) NNs, we sample each minibatch to contain equal numbers of points labelled 0 and labelled 1.

For both the baseline GBDT classifier and the projecting method, we search over a grid of GBDT hyperparameters on ten 80% train/20% test splits, and choose the set of hyperparameters that optimizes the average balanced test accuracy over those ten splits. See C.2 for the parameters that we tune and the code for the values that we consider. For BuDRO, we tune parameters by hand on one 80% train/20% split. We use the dual implementation to find the optimal transport map (without SGD, see Algorithm 7); thus the only extra hyperparameter to consider is the perturbation budget $\epsilon$. The data in Table 1 are collected by averaging the results obtained using the optimal hyperparameter choices across 10 new 80% train/20% test splits (the same splits for each method). The optimal hyperparameters are presented in Table 4.

Table 4: Optimal XGBoost parameters for German credit data set. For BuDRO, we also used a pertubation budget of $\epsilon = 1.0$.

| Method | max_depth | lambda | min_weight | eta | steps |
|---|---|---|---|---|---|
| Baseline | 10 | 1000 | 2 | 0.5 | 105 |
| Projecting | 7 | 2000 | 2 | 0.5 | 111 |
| BuDRO | 4 | 1.0 | 1/80 | 0.005 | 90 |

We also train a neural network on the German credit data set (see Appendix C.2 for a description of the architecture). We find that we consistently obtain high accuracy when we use a learning rate of $10^{-4}$ and run for 4100 epochs (without any $\ell_2$ regularization).

**Results**  Table 5 reproduces the data from the main text (averages over ten 80% train/20% test splits) including standard deviation values.

Table 5: Results on German credit data set. We report the balanced accuracy in the second column. These results are based on 10 splits into 80% training and 20% test data.

| Method | BAcc | Status cons | Age gaps | |
|---|---|---|---|---|
| | | | $\text{GAP}_{\text{Max}}$ | $\text{GAP}_{\text{RMS}}$ |
| BuDRO | 0.715±0.032 | **0.974**±0.025 | **0.185**±0.055 | 0.151±0.048 |
| Baseline | **0.723**±0.019 | 0.920±0.022 | 0.310±0.159 | 0.241±0.109 |
| Project | 0.698±0.024 | 0.960±0.029 | 0.188±0.086 | **0.144**±0.064 |
| Baseline NN | 0.687±0.031 | 0.826±0.028 | 0.234±0.126 | 0.179±0.093 |

## C.4 ADULT

The Adult data set (Dua & Graff, 2017) is another commonly considered benchmark data set in the algorithmic fairness literature. After preprocessing and removing entries that contain missing data, we consider a subset of Adult containing information about 11 demographic attributes from 45222 individuals. Each individual is labelled with a binary label that is 0 if the individual makes less than $50000 per year and 1 if the individual makes more than $50000 per year. In our preprocessing, we standardize the (five) continuous features and one-hot encode the (six) categorical features, resulting in 41 total features that we use in our analysis. These features include several attributes that could be considered protected, such as `age` (continuous), `relationship_status` (categorical), `gender` (binary), and `race` (here we consider a binary white vs non-white `race` feature). For this experiment, we choose to construct a predictor $f$ for the labels that is fair according to the `gender` and `race` attributes.

As mentioned in the main text, we exactly follow the experimental set-up as described in Yurochkin et al. (2020) for the Adult data set. In particular, we do not remove the `gender` and `race` attributes from the training data. This helps to produce an adversarial analysis (how fair can the method become when it is explicitly training on gender and race).

**Fair distance**    The fair distance on Adult is described in detail in C.2. Briefly, we consider three protected directions: $e_g$, the indicator for the `gender` attribute; $e_r$, the indicator for the `race` attribute; and $w$, the normal vector to the separating hyperplane obtained via logistic regression trained to predict the `gender` attribute from the other attributes.

**Evaluation metrics**    As previously mentioned, we consider both gender and race as protected attributes on the Adult data set; thus we report $\text{Gap}_{\text{Max}}$ and $\text{Gap}_{\text{RMS}}$ for both the `gender` and `race` features separately.

We examine two types of counterfactual individual fairness metrics. The first we refer to as *spouse consistency* (S-cons). The S-cons is determined by creating two copies of the data: one in which every point belongs to the `husband` category of the `relationship_status` feature and the other in which every point belongs to the `wife` category. Unlike the framework described in C.2, we do not consider all categories of the `relationship_status` feature. To be concrete, two altered copies of the data are used to obtain two vectors of predicted outcomes $\hat{y}^h$ and $\hat{y}^w$, and the S-cons is the fraction of outcomes that are the same in these two vectors. Explicitly, S-cons is calculated as $\frac{1}{n}|\{i\colon \hat{y}_i^w = \hat{y}_i^h\}|$. Intuitively, the S-cons measures how likely an individual is to be assigned to a different outcome simply from labeling themselves as a `wife` rather than a `husband`.

The other evaluation metric is the *gender and race consistency* (GR-cons), which involves four copies of the input data. Each copy is altered so that the `gender` and `race` features are constant on that copy of the data, and so that each copy has a different combination of the `race` and `gender` feature from the other three copies. We then apply the classifier to each copy of the data, to produce a four vectors of predicted outcomes $\hat{y}_i \in \{0,1\}^n$, $i = 0, 1, 2, 3$. The GR-cons is defined as the fraction of outcomes that are the same in all of the $y_i$.

**Method training and hyperparameter selection**    The labels for the Adult data set are unbalanced (only about 25% of people make more that $50000$ per year) and thus all methods are again trained for balanced accuracy. For the GBDT methods (Baseline, projecting, BuDRO) this is accomplished by setting the XGBoost parameter `scale_pos_weight` to be the ratio of zeros to ones in the labels of the training data. The data from the other methods (baseline NNs, SenSR, and adversarial debaising) are obtained from Yurochkin et al. (2020); see that work for further information about the method training.

The baseline GBDT and projecting methods were trained to optimize balanced test accuracy over a grid of hyperparameters on one 80% train/20% test seed. The optimal hyperparameters discovered in this way were then used to collect data on ten different 80% train/20% test splits: the average of the results on the test data from these new train/test splits are presented in Table 2 in the main text (see Table 7 for information about standard error). The optimal GBDT parameters are presented in Table 6.

Table 6: Optimal XGBoost parameters for the Adult data set. For BuDRO, we also used a perturbation budget of $\epsilon = 0.4$. The value of `min_weight` for BuDRO is computed relative to the size of an 80% training set, which contains 36177 individuals.

| Method | max_depth | lambda | min_weight | eta | steps |
|---|---|---|---|---|---|
| Baseline | 3 | 0.01 | 0.5 | 0.05 | 816 |
| Projecting | 8 | 0.5 | 0.5 | 0.05 | 668 |
| BuDRO | 14 | $10^{-4}$ | 0.1/36177 | 0.005 | 180 |

BuDRO was implemented using the Dual SGD formulation to find the optimal transport map $\Pi$ as described in Algorithm 2. This involves several additional hyperparameters that we set in the following ways: an initial guess of the dual variable (set to 0.1), a batch size (set to 200), a number of SGD iterations (set to 100), a momentum parameter (set to 0.9), and an SGD learning rate (set to $10^{-4}$). The optimal perturbation budget $\epsilon$ was found to be 0.4. See C.4.1 for more information about the hyperparameter selection on Adult.

### C.4.1 FURTHER RESULTS

Table 7 reproduces the results from the main text (averages over 10 80% train/20% test splits) including standard deviations. Only the data collected from GBDT methods are reproduced in Table 7; the data from other methods were obtained from Yurochkin et al. (2020). See Yurochkin et al. (2020) for information about the standard error in these values.

Table 7: Results on Adult. We report the balanced accuracy in the second column.

| Method | Acc | Individual fairness | | Gender gaps | | Race gaps | |
| | | S-cons | GR-cons | GAP$_{\text{Max}}$ | GAP$_{\text{RMS}}$ | GAP$_{\text{Max}}$ | GAP$_{\text{RMS}}$ |
|---|---|---|---|---|---|---|---|
| BuDRO | .815±0.005 | **.944**±0.013 | .957±0.007 | .146±0.012 | .114±0.013 | .083±0.018 | .072±0.017 |
| Baseline | **.844**±0.003 | .942±0.007 | .913±0.008 | .200±0.006 | .166±0.008 | .098±0.013 | .082±0.013 |
| Project | .787±0.005 | .881±0.079 | **1**±0.000 | **.079**±0.022 | .069±0.018 | .064±0.021 | .050±0.016 |
| Reweigh | .784±0.005 | .853±0.010 | .949±0.009 | .131±0.021 | .093±0.015 | **.056**±0.031 | **.043**±0.022 |

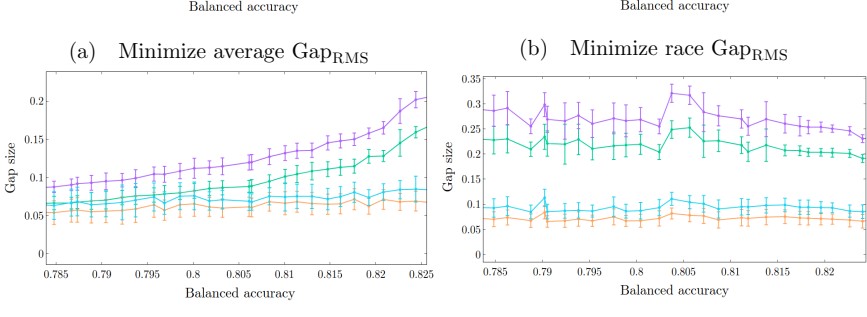

(a)   Minimize average Gap$_{\text{RMS}}$

(b)   Minimize race Gap$_{\text{RMS}}$

(c)   Minimize gender Gap$_{\text{RMS}}$

(d)   Maximize S-cons

Figure 2: Fairness measures at given accuracy levels for the BuDRO method on the Adult data set, considered over 10 train/test splits. The results from each choice of hyperparameters are averaged over all train/test splits before being grouped into accuracy bins. The plotted points are the ones from each bin that optimize the specified quantity ((a) average Gap$_{\text{RMS}}$, (b) race Gap$_{\text{RMS}}$, (c) gender Gap$_{\text{RMS}}$, (d) spouse consistency). Error bars represent one standard deviation. Empirically, the gender gaps were harder to reduce than the race gaps. The S-cons in all of these pictures never drops below 94%.

We illustrate the fairness of BuDRO at different accuracy levels in Figures 2 and 3. The data in Figure 2 was also used to guide hyperparameter selection for the BuDRO method on the Adult data set.

Before analyzing the information in these figures, we provide a description of how they were generated. We separate the accuracy axis (horizontal) into bins of a fixed width (here, the bin size is 0.0016). We then consider ten 80% train/20% test splits of the data set, and explore a grid of hyperparameters for each of these train/test splits (see the included code for the specific hyperparameter grids that we examined). The results from each choice of hyperparameters are averaged (over all ten train/test splits) and are placed in the bin containing the average accuracy; the error bars in the figures correspond to the standard error from this averaging.

In Figure 2, we present four figures: each figure is obtained by determining the set of hyperparameters in each accuracy bin that optimize the average (over the 10 seeds) of a given fairness quantity (on the test set). Thus, each figure contains the data from approximately 30 hyperparameter selections (one for each bin, different for each figure). For example, in Figure 2(c), each point was obtained from the classifier constructed using the hyperparameters that minimize the gender GAP$_{\text{RMS}}$ in

the corresponding accuracy bin. In a similar fashion, Figure 2(a) is constructed by selecting the hyperparameters from each accuracy bin that minimize the average of the race $\text{GAP}_{\text{RMS}}$ and the gender $\text{GAP}_{\text{RMS}}$. In all of the plots, the S-cons never drops below 94%, and the GR-cons remains similarly high; thus, we do not focus on the consistency measures here.

In the following analysis, we concentrate on minimizing the gender $\text{GAP}_{\text{RMS}}$ (Figure 2(c)) due to the fact that the gender gaps were significantly more difficult to shrink than the race gaps in Figure 2. In fact, as seen in Table 2, the baseline classier exhibits fairly small race gaps even though it is trained with knowledge of the `race` feature. In real-world use, however, it is not clear which fairness quantity a user would be required to optimize; thus, Figure 2 presents a picture of the different trade-offs involved.

The BuDRO data in Table 2 was collected from ten different (new) 80% train/20% test splits, using hyperparameters chosen by examining Figure 2(c). We were interested in finding a point with high accuracy and high spouse consistency; thus, we examined hyperparameters corresponding to the points in the accuracy range $0.81$ to $0.825$ and looked for patterns in these hyperparameters. We attempt this generalization to make hyperparameter selection slightly more realistic: it seems willfully ignorant to disregard all of this data when selecting hyperparameters for our final tests, and it is at least plausible that a user could find some of these generally good hyperparameters via hand-tuning. This results in the set of BuDRO hyperparameters that were presented in Table 6. See additionally the code for more details about the selected hyperparameters.

In Figures 2(a) and (c), we observe a trade-off between accuracy and fairness with the BuDRO method. That is, by decreasing accuracy (which generally corresponds to increasing the perturbation parameter $\epsilon$), we are able to decrease the group fairness gaps, all at a high level of S-cons. Thus, it is possible to chose parameters to produce smaller group fairness gaps (with potentially lower accuracy) if the application calls for that.

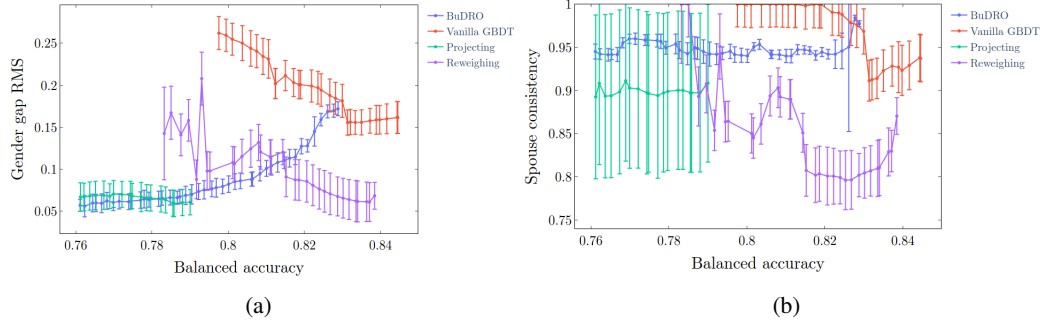

Figure 3: The accuracy vs fairness trade-off of BuDRO when compared to other baseline boosting algorithms on the Adult data set. All lines are chosen to minimize gender $\text{Gap}_{\text{RMS}}$. (a) contains a comparison of the Gender gap RMS. The BuDRO line also appears in Figure 2(c). (b) contains a comparison of the S-cons. Error bars represent one standard deviation.

Figure 3 includes a comparison of the BuDRO method to some other baseline boosting methods to further illustrate the trade-off between accuracy and fairness. Each point in Figure 3 is chosen to be the one from the accuracy bin that minimizes the gender $\text{Gap}_{\text{RMS}}$.

This figure is constructed specifically to explore how fair we can make a classifier at given (fixed) levels of accuracy. The vanilla GBDT method always produces high accuracy classifiers (in the hyperparameter grid that we consider). This comes with large group fairness gaps that we are unable to decrease. On the other hand, the projecting method always produces good group fairness gaps, but we are unable to obtain high accuracy with this method. Finally, the reweighing method can obtain high accuracy with low fairness gaps, but it never produces an acceptable S-cons.

Figure 3 suggests that BuDRO is a better method than projecting at all accuracy levels on Adult: at any accuracy level, BuDRO matches the group fairness gaps produced by projecting while improving upon the (consistency of the) individual fairness metric. Projecting always produces a classifier with small group fairness gaps. Unlike projecting, however, BuDRO can also be used to construct a more accurate classifier if we are allow for slightly larger group fairness gaps. Depending on the

requirements of the application (e.g. as defined by law or other application specific fairness goals), this allows for more flexibility in the creation of individually fair and accurate classifiers.

It is also interesting to observe how the BuDRO curve in Figure 3 meets the curve for Vanilla GBDTs. Specifically, as the curves meet, the gender gap of the BuDRO method is increasing, while the gender gap of the Vanilla GBDT curve appears to be slightly decreasing. We speculate that this is due to the fact that the grid of hyperparameters used in the construction of Figure 3 does not contain values of $\epsilon$ smaller that 0.1. That is, the value of $\epsilon$ does not go down to 0, so we can not expect to precisely recover the baseline results. Essentially, we *force* a small perturbation in the data while also pushing for high accuracy. Thus, it appears that the high accuracy points in the Figure correspond to solutions that are obtained by improving race fairness without significantly improving gender fairness (see also Figure 2(c) - the race fairness always remains small). These high accuracy cases apparently find a perturbation that is mostly along the race axis.

Overall, these figures provide further evidence that the BuDRO method creates an individually fair classifier while still obtaining high accuracy (on tabular, structured data like Adult) due to the ability to leverage powerful GBDT methods.

## C.5   COMPAS

The COMPAS recidivism prediction data set, compiled by ProPublica (Larson et al., 2016), includes information on the offenders' `gender`, `race`, `age`, criminal history (`charge_for_arrest`, `number_prior_offenses`), and the `risk_score` assigned to the offender by COMPAS. ProPublica also collects whether or not these offenders recidivate within two years as the ground truth. More details and discussions of the data set can be found in Angwin & Larson (2016); Flores et al. (2016).

We remove the `risk_score` attribute, standardize the `number_prior_offenses` attribute and one-hot encode the `age` attribute, yielding the final data set of 5278 individuals with 7 features.

Since there are only seven features (with only one continuous feature) in the COMPAS data, there are many different pairs of individuals $(i, j)$ with the same similarity distance $C_{i,j}$. Then, there can be a lot of different solutions while finding $t \in \arg\max_i R_{ij} - \eta^* C_{ij}$ when using the dual method (see Appendix B.2 and Algorithm 7). Therefore, we consider the entropic regularization form (B.1) of the linear program (2.8), and solve it using the fast Sinkhorn method in Algorithm 4 and 5.

**Fair metric**   We define three protected directions: $e_g$, the indicator vector of `gender`, $e_r$, the indicator vector of `race`, and a protected direction $w$ obtained by eliminating the `race` attribute and training the logistic regression for binary label `race` on the edited data set. Then, we follow the steps for computing the projection matrix $A = \text{span}\{e_g, e_r, w\}$ and calculating the fair distances $d_x$ in (C.1) for all pairs of individuals.

**Evaluation metrics**   We evaluate the methods using several individual fairness metrics, as well as some group fairness metrics. For group fairness metrics, we report $\text{GAP}_{\text{Max}}$ and $\text{GAP}_{\text{RMS}}$ for `race` and `gender` separately. We consider two counterfactual individual fairness evaluation measures on COMPAS based on the two protected attributes `race` and `gender` (which are both binary). Therefore, we can make the counterfactual examples by flipping the protected attributes. For example, we generate two copies of the data set: one with all individuals are female, the other with all individuals are male. Then the *gender consistency* (G-cons) is obtained by calculating the fraction of the same classified outcomes on two copies. The *race consistency* (R-cons) can be calculated in a similar way.

**Method training and hyperparameter selection**   We compare to the same methods that were considered on Adult: baseline GBDT, projecting, and reweighing (based on XGBoost); and baseline NN, SenSR, and adversarial debiasing (based on neural networks).

The results reported are averaged over ten random training (80%) and testing (20%) splits of the data set. We implement the adversarial debiasing methods (in the `adversarial_debiasing` class from the IBM's `AIF360` package, version `0.2.2` (Bellamy et al., 2018)) with the default hyperparameters, and combine the implementation of reweighing (in the `reweighing` class from `AIF360` with default hyperparameters) with running XGBoost (default parameters) for 100 boosting

steps. For the baseline GBDT and projecting, we select hyperparameters by splitting 20% of the training data into a validation set and evaluating the performance on the validation set. For Baseline NN, SenSR and the BuDRO methods, we manually train the hyperparameters on one train/test split and use the resulting hyperparameters for computing the averaged results of 10 restarts.

We report the optimal hyperparameters for the GBDT methods in Table 8. For these methods, `lambda` is $10^{-8}$, and `min_child_weight` is $0.1/n_{\text{train}}$, where $n_{\text{train}}$ is the number of training samples. We set `scale_pos_weight` to 1 for projecting and the baseline, `scale_pos_weight` to be the ratio of zeros to ones in the labels of the training data for BuDRO. The optimal perturbation budget $\epsilon$ for BuDRO is 0.12.

Table 8: Optimal XGBoost parameters for COMPAS data set.

| Method | max_depth | eta | steps |
|---|---|---|---|
| Baseline | 3 | $5 \times 10^{-4}$ | 1600 |
| Projecting | 4 | $7.5 \times 10^{-4}$ | 2000 |
| BuDRO | 2 | $1.5 \times 10^{-5}$ | 68 |

For the neural network based methods, the optimal hyperparameters are described below. For baseline NN: learning rate = $5 \times 10^{-6}$, number of epochs = 27000. For SenSR: perturbation budget = 0.01, epoch = 2000, full epoch = 10, full learning rate = 0.0001, subspace epoch = 10, subspace learning rate = 0.1.

**Results** The results in Table 3 with the associated standard deviations of BuDRO and the comparison methods are shown in Table 9.

Table 9: Results on COMPAS data set. These results are based on 10 random splits into 80% training and 20% test data.

| Method | Acc | Individual fairness | | Gender gaps | | Race gaps | |
|---|---|---|---|---|---|---|---|
| | | G-cons | R-cons | $\text{GAP}_{\text{Max}}$ | $\text{GAP}_{\text{RMS}}$ | $\text{GAP}_{\text{Max}}$ | $\text{GAP}_{\text{RMS}}$ |
| BuDRO | 0.652±0.012 | **1.000**±0.000 | **1.000**±0.000 | **0.124**±0.053 | **0.099**±0.037 | 0.145±0.028 | 0.125±0.021 |
| Baseline | 0.677±0.014 | 0.944±0.038 | 0.981±0.040 | 0.223±0.055 | 0.180±0.041 | 0.258±0.056 | 0.215±0.046 |
| Project | 0.671±0.012 | 0.873±0.025 | 1.000±0.000 | 0.189±0.040 | 0.150±0.030 | 0.230±0.043 | 0.185±0.033 |
| Reweigh | 0.666±0.020 | 0.788±0.023 | 0.813±0.047 | 0.245±0.054 | 0.207±0.046 | **0.092**±0.027 | **0.069**±0.019 |
| Baseline NN | **0.682**±0.011 | 0.841±0.027 | 0.907±0.027 | 0.282±0.045 | 0.246±0.028 | 0.258±0.055 | 0.228±0.047 |
| SenSR | 0.652±0.018 | 0.977±0.029 | 0.988±0.016 | 0.167±0.065 | 0.129±0.047 | 0.179±0.039 | 0.159±0.032 |
| Adv. Deb. | 0.671±0.016 | 0.854±0.028 | 0.818±0.088 | 0.246±0.064 | 0.219±0.051 | 0.130±0.065 | 0.108±0.059 |

## C.6 Method timing information

Table 10 contains the training runtimes of the vanilla GBDT and the BuDRO methods. The runtimes of the other fair GBDT methods (projecting and reweighing) are dominated by the time required for running vanilla GBDT after preprocessing; thus, the runtimes for these methods are omitted here. The results show that the BuDRO method as defined in Algorithm 3 (i.e. using SGD to find a dual solution with $O(n^2)$ time required to recover the primal solution) is scalable to large problems.

Table 10: Average runtime for training, including standard deviations. The number in parentheses is the number of trials used to compute the average. In all trials, the hyperparameters (including the number of boosting steps) were examined during the generation of the data presented in Section 5. Each baseline GBDT trial ran for 1000 boosting steps on 4 CPUs. BuDRO ran for 500 steps on 2 CPUs on German credit, 200 steps using 2 CPUs on COMPAS, and 200 steps using 4 CPUs and 1 GPU on Adult.

| | Problem size | | Time (seconds) | |
|---|---|---|---|---|
| Data set | training samples | features | Baseline GBDT | BuDRO |
| German credit | 800 | 62 | $6.5 \pm 0.3$ (10) | $105.0 \pm 25.5$ (480) |
| COMPAS | 4222 | 7 | $17.5 \pm 0.7$ (10) | $154.9 \pm 4.5$ (10) |
| Adult | 36177 | 41 | $201.6 \pm 5.4$ (10) | $1455.9 \pm 263.2$ (6) |

