# OpenReview forum: "Individually Fair Gradient Boosting"
_ICLR.cc/2021/Conference — ICLR 2021 Spotlight_

### Official Review · AnonReviewer3 · 2020-10-27
**Good paper**

**Rating:** 7
**Confidence:** 4

**Review:**

The authors presented in the submission a thorough study on enforcing the aggregated individual fairness with non-differentiable ML models. The proposed method generates individually fair and robust ML models in a minimax fashion among all possible samples that are close to the true distribution w.r.t. a given fair metric. They introduce the augmented support and transfer the standard gradient descent to a gradient descent in the functional space anchored by the augmented support to optimize the adversarial risk function with non-smooth ML models, e.g. decision trees. Solid theoretical guarantees and convincing empirical study results are provided to support their claims. The paper is highly completed, well-structured (though a bit dense given the page limit) and well-written - a clear accept.

The only major issue in my opinion is that the submission lacks the formal definition and discussion regarding "individual fairness", which is defined on the second page as "f(x_1) \approx f(x_2) if d_x(x_1, x_2) is small" for a classifier "f: X \to {0,1}". This \approx is unclear under this binary classification setting, and the definition leaves me with the impression that it means two close examples by the fair metric should have similar predictions.

However such understanding seems problematic and inaccurate throughout the rest of the paper. Decision trees are almost certain to have a value jump near the boundary of any node, therefore any quantification of the definition (e.g. I am thinking of a \epsilon-\delta argument that |f(x_1)-f(x_2)| < \delta when d_x(x_1, x_2) < \epsilon for fixed \epsilon and \delta) above has to deal with the case when d_x(x_1, x_2) is small but x_1 and x_2 are in two nodes in any single tree in the GBDT ensemble. Also, the restricted empirical adversarial cost function (2.4) appears more like a robust cost function that guarantees the model accuracy for all possible data generating process similar to the truth (which is the d_x part), whereas it has no reference to "f(x_1) \approx f(x_2)".

After (2.4) I feel the individual fairness in the paper might actually mean that "two close examples by the fair metric should be equally accurate", which seems to make more sense given the transportation between similar examples and the risk being the sole optimization target. If this is the case, it might be worth pointing it out at the beginning of the section.

Other than that, it is a very smooth experience of reading the paper.

Minor and editorial issues:

1.  Though stated by the authors that "our method readily extends to other supervised learning setting", the theoretical discussion and the empirical study covered in the paper are both based on solving a binary classification problem.

2. Assumption A.1 requires the loss $l$ is convex, while Assumption 3.2 requires $l$ to be bounded. These two points together make the loss constant unless, e.g., f(x) is uniformly bounded for all f \in \mathcal{F}. In such case, to achieve convergence and generalizability simultaneously, GBDTs might need to be bounded, which undermines Assumption A.2. The interaction between the assumptions seems non trivial and is worth its own discussion.

3. Page 14, Proof of Theorem, "Under Assumptions A.1(iii)-(v)". Assumptions A.1 (iii) - (v) seem missing?

---

> ### Author Response · Authors · 2020-11-19
> **Response to Reviewer 3**
>
> We thank the reviewer for the feedback. We have added the precise notion of individual fairness to the paper, clarified the multi-class extension and assumptions as you suggested. Please see our detailed answers below.
>
> **The only major issue in my opinion is that the submission lacks the formal definition and discussion regarding "individual fairness". After (2.4) I feel the individual fairness in the paper might actually mean that "two close examples by the fair metric should be equally accurate", which seems to make more sense given the transportation between similar examples and the risk being the sole optimization target.**
>
> You are right in pointing that we are using a risk-based notion of individual fairness. This is not a new notion; it first appeared in Yurochkin et al (2020). We added a comment and Definition 2.1 to Section 2 clarifying the precise notion of individual fairness we are enforcing.
>
> **Though stated by the authors that "our method readily extends to other supervised learning setting", the theoretical discussion and the empirical study covered in the paper are both based on solving a binary classification problem.**
>
> We've clarified that it is possible to extend our method to multi-class classification problems, but not problems with a continuous target variable in Section 2.
>
> **Clarification regarding assumptions**
>
> We have in mind a bounded sample space (see Assumption 3.1), so boundedness and convexity of the loss on the sample space is less restrictive. We've clarified this point in Appendix A (immediately following the statement of Assumption A.1).
>
> **Assumptions references**
>
> We've corrected the references to the assumptions in the proof of Theorem 3.4.

---

### Official Review · AnonReviewer2 · 2020-10-28
**Novel and interesting approach for training "individually fair" boosted DTs, but scalability is a concern**

**Rating:** 7
**Confidence:** 4

**Review:**

### POST-REVISION

Thanks for the revisions made to the paper, particularly for elaborating on the heuristic to speed-up the inner subroutine, and the clarification on the generalization bound.

Also thanks for the running time numbers. Would be great if you could report them in the paper (if you haven't already).

I'm revising my score for the paper to 7.
**********

The paper presents an interesting idea to train boosted decision trees to satisfy individual fairness constraints (for a pre-specified similarity metric "d_x"). The prior gradient-based method of Yurochkin et al. for individual fairness constraints does not apply to non-continuous (or non-smooth) models such as decision trees. The authors borrow the same distributionally robust loss setup as Yurochkin et al, where the goal is to minimize classification performance over all distributions that are close to empirical distribution, with the closeness measured in terms of the similarity metric "d_x" (via the optimal transport distance). However, unlike the prior method which works with the dual formulation, the authors show how one solve the primal problem directly with functional gradient descent as long as the "supremum" over distributions in robust loss function can be computed efficiently. They then show how the supremum can be computed using an LP. Interestingly, by directly computing the supremum, the resulting gradient boosted training does not require the model to be continuous in its inputs. The authors provide generalization bounds and experiments on three benchmark fairness datasets.

Pros:
- Training individually fair models is still a nascent area of research, and so the paper makes a valuable contribution to this subarea of ML fairness.
- The trick the authors use to accommodate non-continuous models by directly applying functional gradient descent to the distributionally robust loss function is pretty neat.

Cons:
- The main concern is the scalability of the approach. Each iteration of the gradient boosted training requires solving an LP with n^2 variables, where "n" is the number of training points. The authors provide a solver for the LP (based on some practical tricks) which requires O(n^2) computation, but even this does not seem scalable for large problems.
- I think the main difficulty in scaling up the proposal is having to perform operations on n x n matrices to compute the supremum (e.g. R, C, \Pi in Alg 2). But I guess in practice, you may be able to reduce the storage costs, exploit sparsity patterns (via the proposed regularization) and avoid having to visit all the matrix entries (e.g. using the fact that you only care about examples within a minibatch while performing SGD).  Would be great if you could provide some run-time numbers for your approach for different dataset sizes.
- The generalization bound has an exponential dependence on the dimension "d". The authors say this is unavoidable because of the  relaxation they introduce to the original robust loss where they replace the original dataset with an augmented dataset. How important is the use of the augmented dataset "D_0" to the working of the gradient boosted training (the exponential dependence on "d" seems like a heavy price to pay to enable this relaxation)? I understand that this was needed to ensure that you only need to compute gradients for the loss \ell w.r.t. the scores f(x) and not w.r.t. the labels "y", but I wonder if there's any another way to handle the labels without worsening the generalization bound.

Finally, I have one other questions about the generalization bound Thm 3.4:
- In your proof, you seem to make use of the dual formulation for the robust loss L. But would you need the model to be continuous in its inputs for strong duality to hold? In other words, looking at proposition 1 in Sinha et al. (2018), the equality between the primal and dual objectives seem to require some form of continuity assumption on the loss "l" as a function of "x". Indeed Assumption 3.2 in your paper does assume the loss is Lipschitz continuous but wouldn't this require the model to be continuous in its inputs?

Other comments/questions:
- Sec 2.2: I thought the joint distribution was over "D_0 × D_0" from the previous subsection, but you mention "D_0 x D" here. Is "D" the un-augmented training distribution?
- Convergence of functional gradient descent (Appendix A.1): I think assumption A.2 assumes that the base learners are rich enough to model a close approximation to the gradients ∇L(f). I'm not too familiar with the analysis of gradient boosted DTs, but is there an interpretation of gradient boosting where line 6 in Alg 1 is seen as a "projection step", so that you don't have to make a strong assumption about the base learners? In other words, can gradient boosting be seen as a form of functional gradient descent with a projection step (the only difficulty I guess is that the projection happens before the update step in line 7)
- Proof of Theorem 3.4 in Appendix A.2: Does "L_f" actually mean "L_e"? What does the notation \pi_i(D) denote (I might have missed this from earlier in the appendix)?
- Typo in page 3, para 1: y_1 -> y_i?

---

> ### Author Response · Authors · 2020-11-19
> **Response to Reviewer 2**
>
> Thank you for the feedback. We address questions and concerns below. We have updated the draft accordingly - specific changes are mentioned in the response.
>
>
> **The main concern is the scalability of the approach.**
>
> As you noted, solving eq 2.7 directly (an LP with $n^2$ variables), would be computationally prohibitive for even moderately sized data. To alleviate this issue, in Section 4 (see Appendix B for details) and Algorithm 2 we derived a stochastic optimization for the (scalar) dual variable. Steps 2-7 of Algorithm 2 are quite fast to converge and support mini-batching. The $n^2$ cost comes from a naive implementation of recovering the primal solution (steps 8-11 of Algorithm 2). In practice, it is possible to speed up the recovery of the primal solution by observing that the effective domain of the search in step 10 of Algorithm 2 is only over a small number of points that are close to $x_j$ in the fair metric. This can be seen from the corresponding argmax expression: if $C_{ij}$ is large, $i$ is unlikely to be a solution to argmax. The neighbors of each point can be computed in advance and re-used during training. This reduces the cost of recovering the primal solution from $O(n^2)$ to $O(nm)$, where $m$ is the maximum number of neighbors of a point. With this heuristic it should be possible to train our algorithm whenever the vanilla GBDT is feasible. We have added a comment describing this heuristic to Section 4.
>
> **Would be great if you could provide some run-time numbers for your approach for different dataset sizes.**
>
> Training BuDRO on Adult (36k training points) takes about 24 minutes; vanilla GBDT takes about 3 minutes. On German (800 training points) BuDRO takes 105 seconds; vanilla GBDT takes 6 seconds. On COMPAS (4200 training points) BuDRO takes 155 seconds; vanilla GBDT takes 17 seconds.
>
> **Exponential dependence on the dimension in the generalization error bound.**
>
> This dependence is generally pessimistic. We elaborate on this in the second paragraph following the statement of Theorem 3.4. At a high level, the exponential dependence on dimension is pessimistic because the fair metric restricts the dimension of the search space for differential performance to a small "similar neighborhoods" of the sample space. This intuition is confirmed by the fact that all the fairness metrics we report in section 5 are on a held-out test set; i.e. individual fairness during training generalizes.
>
> **Continuity of the model for strong duality.**
>
> The assumption of continuity of the model in Sinha et al. (2018) is needed in the argument for strong duality to permit an interchange of maximization and expectation. In other words, it is a technical assumption made to simplify the proof. It is possible to relax this assumption to accommodate models with jumps by appealing to more general strong duality results (e.g. see Blanchet and Murthy (2016) Thm 1). We updated the references in the proof of Thm 3.4 to reflect the need to invoke this more general duality result.
>
> **Other questions:**
>
> * $\mathcal{D}$ is the unaugmented training distribution, and we write $z_j\in\mathcal{D}$ because the second argument to the cost function is always an original training example in the BuDRO objective function. However, it is just as correct to write $\mathcal{D}_0$ in lieu of $\mathcal{D}$ here because $\mathcal{D}\subset\mathcal{D}_0$ (and $P_n$ is only supported on $\mathcal{D}$).
> * You are correct in noting the difference between gradient boosting and projected gradient descent: in gradient boosting, the projection step occurs before the update.  The assumption on the richness of the base learners is usually satisfied in practice (e.g. decision trees are universal approximators), so this assumption is common in the literature on gradient boosting.
> * $L_e$ and $L_f$ are distinct. Intuitively, $L_e$ is the objective function in which we don't restrict the search to the augmented dataset; we search over the entire sample space. This is the objective that prior works on training individually fair ML models use. The $\pi_1(\mathcal{D})$ notation appeared in a previous draft; this notation has been replaced with more standard notation.
> * We fixed this typo in the revised version.

---

### Official Review · AnonReviewer4 · 2020-10-29
**Recommendation to Accept**

**Rating:** 7
**Confidence:** 2

**Review:**

This paper proposes a non-smooth method to enforce individual fairness in gradient boosting. To deal with the non-smoothness of the model, it restricts the optimal transport distance to that defined on an augmented training support set and thus reduces the search of a worst-case distribution to solving an LP problem, where an approximate solution can be found efficiently by SGD on the dual space. The authors provide convergence and generalization properties of the algorithm, and demonstrate its improvement of group and individual fairness metrics in several numerical experiments.

The paper is well established and written. The proposed method is novel and the experiment results look promising.

Comments and Questions:
1. It seems that this method can be readily extended to multi-class classification problems. Does this method work with continuous output space? What would be the augmented set like?
2. The definition of W_D above (2.4) looks confusing as it is exactly the same as W. We may want to highlight the distinction as it is defined on the augmented support set.
3. A typo in (2.5) where there is no summation over i?
4. The definition of R is confusing, is it just \ell(f(x_i), y_j)?
5. How long does it take to run BuDRO on the three experiments, and compared to other methods?

---

> ### Author Response · Authors · 2020-11-19
> **Response Reviewer 4**
>
> Thank you for the feedback. We have updated the draft according to your suggestions and address the questions below.
>
> 1. It is unclear how to extend this approach to continuous output spaces. Following the approach in the paper, the augmented set would be $\\{x_1\times\mathcal{Y},\dots,x_n\times\mathcal{Y}\\}$, but this is not a set that we can easily optimize over. We've clarified that it is possible to extend our method to multi-class classification problems, but not problems with a continuous target variable at the beginning of Section 2.
> 2. We've clarified the feasible set in the definition of $W_{\mathcal{D}}$.
> 3. There is no summation over $i$ because we are writing out $\frac{\partial L}{\partial\hat{y}_i}$ in (2.5)
> 4. We have clarified the definition of $R$ in Algorithm 1
> 5. Training BuDRO on Adult (36k training points) takes about 24 minutes; vanilla GBDT takes about 3 minutes. On German (800 training points) BuDRO takes 105 seconds; vanilla GBDT takes 6 seconds. On COMPAS (4200 training points) BuDRO takes 155 seconds; vanilla GBDT takes 17 seconds. Run-times of the pre-processing methods (Project and Reweigh) are similar to vanilla GBDT, but they are not effective as shown in our experiments. Please also see our response to Reviewer 2 for a discussion regarding the scalability of our method. We have added a paragraph to section 4 describing a heuristic for achieving further speedups.

---

### Author Response · Authors · 2020-11-19
**General response**

We thank all the reviewers for the thoughtful comments. We answer each reviewer’s questions individually and we have updated the draft according to the feedback.

---

### Decision · Program_Chairs · 2021-01-07
**Final Decision**

**Decision:**

Accept (Spotlight)

**Comment:**

The paper provides a method to train boosted decision trees to satisfy individual fairness. All of the reviews suggest that this paper is well-written and gives novel techniques for solving an interesting problem. The authors have addressed most of the concerns raised by the reviewers during their response. However, the authors should follow a suggestion in the reviews and include the running time in the empirical evaluation.